# Crude metabolites from endophytic fungi inhabiting Cameroonian *Annona muricata* inhibit the causative agents of urinary tract infections

Lorette Victorine Yimgang, Rufin Marie Kouipou Toghueo*, Ines Michele Kanko Mbekou, Darline Dize, Fabrice Fekam Boyom*

Antimicrobial & Biocontrol Agents Unit (AmBcAU), Laboratory for Phytobiochemistry and Medicinal Plants Studies, Department of Biochemistry, Faculty of Science, University of Yaoundé I, Cameroon, Messa, Yaoundé, Cameroon

* toghueo.rufin@yahoo.fr (RMKT); fabrice.boyom@fulbrightmail.org (FFB)

**Data Availability Statement:** All relevant data are within the manuscript and its Supporting Information files.

## Abstract

Urinary tract infections (UTIs) are common bacterial infections. The global emergence of multidrug-resistant uropathogens in the last decade underlines the need to search for new antibiotics with novel mechanisms of action. In this regard, exploring endophytic fungi inhabiting medicinal plants used locally against urinary tract infections could be a promising strategy for novel drug discovery. This study investigates crude metabolites from endophytic fungi isolated from *Annona muricata* as potential sources of antibiotic drugs to fight against uropathogens and reduce related oxidative stress. Crude ethyl acetate extracts from 41 different endophytic fungi were screened against three bacterial strains using the broth microdilution method, and fungi producing active crude extracts were identified using ITS1-5.8S rRNA-ITS2 nucleotide sequences. The antibacterial modes of action of the five most active extracts were evaluated using *Staphylococcus aureus* ATCC 43300 and *Klebsiella oxytoca* strains. The DPPH and FRAP assays were used to investigate their antioxidant activity, and their cytotoxicity against the Vero cell line was evaluated using the MTT assay. Out of the 41 crude extracts tested, 17 were active with minimum inhibitory concentrations (MICs) ranging from 3.125 μg/mL to 100 μg/mL and were not cytotoxic against Vero cell lines with a cytotoxic concentration 50 ($CC_{50}$) >100 μg/mL. The more potent extracts (from *Fusarium waltergamsii* AMtw3, *Aspergillus* sp. AMtf15, *Penicillium citrinum* AMf6, *Curvularia* sp. AMf4, and *Talaromyces annesophieae* AMsb23) significantly inhibited bacterial catalase activity, lysed bacterial cells, increased outer membrane permeability, and inhibited biofilm formation, and the time-kill kinetic assay revealed concentration-dependent bactericidal activity. All seventeen extracts showed weak ferric iron-reducing power (1.06 to 12.37 μg equivalent $NH_2OH$/ g of extract). In comparison, seven extracts exhibited DPPH free radical scavenging activity, with $RSA_{50}$ ranging from 146.05 to 799.75 μg/mL. The molecular identification of the seventeen active fungi revealed that they belong to six distinct genera, including *Aspergillus*, *Curvularia*, *Fusarium*, *Meyerozyma*, *Penicillium*, and *Talaromyces*. This investigation demonstrated that fungal endophytes from Cameroonian *Annona muricata*, a medicinal

**Funding:** The author(s) received no specific funding for this work.

**Competing interests:** The authors have declared that no competing interests exist.

**Abbreviations:** AMr, *Annona muricata* root; AMrb, Annona muricata root bark; AMf, *Annona muricata* fruit; AMs, *Annona muricata* seed; AMtf, *Annona muricata* thorn of fruit; AMtw, *Annona muricata* twigs; AMl, *Annona muricata* leaves; AMb, *Annona muricata* bark; AMsb, *Annona muricata* stem bark; AMpe, *Annona muricata* peducle; ATCC, American Type Culture Collection used as reference strains for respective bacteria; NB, Nutrient Both; CPC, Centre Pasteur du Cameroun; CLSI, Clinical Laboratory Standard Institute; PDA, Potato dextrose agar; PDB, Potato dextrose broth; MIC, Minimal Inhibitory Concentration; BLAST, Basic Local Alignment Search Tool; CBS, Centraalbureau voor Schimmelcultures; $CC_{50}$, 50% cell cytotoxic concentration; DMEM, Dulbecco's minimum essential medium; PCR, Polymerase Chain Reaction; ITS, Iternal Transcribed Spacer; MEGA, Molecular Evolutionary Genetics Analysis; PCR, Polymerase Chain Reaction; NA, Not applicable; SD, Standard deviation; HNC, Herbier National du Cameroun; NCBI, National Center for Biotechnology Information; MTT, 3-(4,5-dimethylthiazol-2-yl)-2,5-diphenyl tetrazolium; FRAP, Ferric Ion Reducing Antioxidant Power; DPPH, 2,2-diphenyl-1-picryl-hydrazyl; $RSA_{50}$, 50% Radical Scavenging Activity; $EC_{50}$, 50% efficient concentration; ARP, Antiradical Power; EDTA, Ethylene Diamine Tetraacetic Acid-Na2; DNA, Deoxyribonucleic Acid.

plant used locally to treat bacterial infections, might contain potent antibacterial metabolites with multiple modes of action. The antibacterial-guided fractionation of these active extracts is currently ongoing to purify and characterise potential antibacterial active ingredients.

## Introduction

Infectious diseases emerging throughout history have included some of the most feared plagues of the past. Recent outbreaks include Ebola, Zika, dengue, Middle East respiratory syndrome (MERS), severe acute respiratory syndrome (SARS), influenza, fungal infection, and more recently, COVID-19 [1, 2]. Although new infections emerging today are global problems, many old diseases and the looming specter of rising antimicrobial resistance are still with us. They are continuously jeopardising human health and various forms of social and economic well-being [1]. In this respect, urinary tract infections (UTIs) are among the most common bacterial infections affecting individuals of all ages and sexes [3]. Currently, it represents a serious public health problem, causing approximately 150 million cases per annum [4, 5], with an estimated annual cost of $1.6 billion in the United States alone [6]. In Cameroon, a recent investigation revealed a prevalence of 12% among children from 1 to 3 years of age at the Buea District Hospital [7].

Testing and treatment strategies vary for the management of UTIs, with empirical therapy being the use of broad-spectrum antibiotics [4, 8]. Unfortunately, in recent years, we have seen the utility of many of these antibiotic agents eroded because of their widespread use and the subsequent development of resistance [9–11]. According to the World Health Organization (WHO), antibiotic resistance remains one of the greatest threats to human health and poses a severe financial burden on healthcare systems worldwide [12]. The increase in resistance to many of the currently available oral drugs makes the management of UTIs caused by resistant pathogens a significant challenge. Therefore, there is an urgency to develop new effective antibiotics with a novel mechanism of action to overcome antibiotic resistance.

Natural products are the primary source of antibiotics, most of which are produced by microorganisms. Microorganisms, particularly fungi, represent a large and still resourceful pool for discovering novel compounds to combat antibiotic resistance in human pathogens. Their ability to produce structurally diverse and potent compounds has been known for decades, particularly with the discovery of penicillin G, a β-lactam antibiotic from *Penicillium notatum* by Sir Alexander Fleming [13–15]. Over the past decades, endophytic fungi living in a symbiotic association with medicinal plants without causing harm have been recognized as excellent sources of structurally novel and bioactive antimicrobial agents [12, 15, 16]. The Cameroonian biodiversity is rich, with thousands of plant species often used by local populations to treat various infections. In fact, with over 4,000 plant species per degree squared, Cameroon's plant diversity is higher than that of all other West African countries combined [17]. Therefore, the long-neglected microbiome of these plants deserves serious attention. Their investigation could hold a key for future great discoveries in the fight against antibiotic-resistant microbes and urinary tract infections.

In our continuous bioprospecting of endophytic fungi residing inside Cameroonian medicinal plants as sources of potent antimicrobial agents, we screened endophytic fungi from *Annona muricata* for their potential antibacterial activity. Our previous investigation revealed that these endophytes could produce crude metabolites with antiplasmodial activity [18]. *Annona muricata* is a multipurpose plant traditionally used in Cameroon and elsewhere to

treat several infectious diseases, including UTIs [19]. We hypothesized that these endophytic fungi might produce active metabolites against bacterial pathogens that are causative agents of UTIs. The present study investigates the antibacterial and modes of action of endophytic fungal extracts isolated from the plant mentioned above.

## Materials and methods

### Source of endophytic fungi

The forty-one endophytic fungi used in this study were isolated from healthy and mature organs (root, root bark, fruit, the thorn of fruit, seed, twig, leaf, stem bark, bark, and peduncle) of *Annona muricata* (3289/HNC). All plant samples were collected in Yaoundé, Cameroon, on January 10, 2016. Small pieces of plant materials measuring approximately 2 mm were surface disinfected through a 5-min rinse with 70% ethanol, followed by treatment with 1% active chlorine solution for 15 min, 2 min in 70% ethanol, and a final rinse three times in sterile distilled water. Disinfected material pieces were plated on potato dextrose agar (PDA; HiMEDIA, India) containing chloramphenicol (200 mg/L) and kept in the dark at room temperature (22–26˚C). After the emergence of endophyte mycelium from plant tissues into the agar, mycelial fragments were transferred to fresh PDA plates and maintained under natural light at room temperature. All endophytes were kept at -80˚C in a 50% glycerol solution at the Antimicrobial & Biocontrol Agents Unit (AmBcAU), Laboratory for Phytobiochemistry and Medicinal Plants Studies, Department of Biochemistry, Faculty of Science, University of Yaoundé I, Cameroon[18].

### Molecular identification of potent antibacterial endophytes

The endophytic fungal isolates showing antibacterial activity against UTI pathogens at 100 μg/mL were identified based on the ITS1-5.8S rRNA-ITS2 nucleotide sequence. Briefly, fungal genomic DNA was extracted from mycelium grown in potato dextrose broth (PDB, HIMEDIA) using a commercial kit (RedExtract-N-Amp Plant PCR, Sigma Aldrich, USA). The extracted DNA concentration and purity (A260/A280 ratio) were measured with a Thermo Scientific NanoDrop 1000 Spectrophotometer (Thermo Scientific, Germany) using 1 μL of each sample. The ITS1-5.8S rRNA-ITS2 region was amplified by PCR using primers ITS4 and ITS5 (Sigma–Aldrich, Germany) and the protocol described by White *et al.* [20]. The reagents included in the kit were also used for PCR amplification using the following conditions: 95˚C for 2 min, followed by 35 cycles of 94˚C for 1 min, 54˚C for 1 min, and 72˚C for 1 min; and a final step of 72˚C for 10 min. The PCR product was analyzed by agarose (1% agarose) gel electrophoresis using 1×Tris EDTA (TE) buffer containing 1 μg/mL ethidium bromide (EtBr) and a constant voltage of 100 V for 20 min. DNA bands were visualized, and images were acquired using a Gel Doc XR+ imaging system (Bio–Rad Laboratories Inc., Germany). Amplicons were purified by filtration (MSB Spin PCRapace, Invitek, Germany), and only one strand of the PCR amplicon was sequenced. The sequence reaction was started at the 5′ end of the ITS1-5,8S rRNA-ITS2 region using the primer ITS4.

The BLAST algorithm was used to find sequences similar to those obtained from fungal isolates. The criteria for identifying isolates were based on the similarity of their sequences to those of reliable reference isolates included in open access nucleotide databases. A dendrogram was made with the nucleotide sequences of the isolates and those of reference strains deposited in Centraalbureau Voor Schimmelcultures (CBS) and American Type Culture Collection (ATCC) collections. Sequences were aligned using the following parameters: pairwise alignment parameters (gap opening = 10 and gap extension = 0.1) and multiple alignment parameters (gap opening = 10, gap extension = 0.2, transition weight = 0.5, and delay divergent

sequences = 25%) and optimized manually in MEGA 7.0. For the phylogenetic analyses based on maximum likelihood (ML), the best-fit models of nucleotide substitution for each data partition were determined using MEGA 7.0 software and incorporated into the analyses. Alignment gaps were treated as partial missing information, and one thousand replications estimated the robustness of the classifications. The initial trees for the heuristic ML search were obtained by applying the neighbor-joining method to a matrix of pairwise distances estimated using the maximum composite likelihood approach, allowing some sites to be evolutionarily invariable. Groups of sequences at proximity within the same branch of the dendrogram were individually aligned to determine their similarity percentage. Sequences with close similarity with reference sequences used for phylogenetic analysis were considered to belong to the same species as the reference sequence [21].

## Culture and extraction of endophytic fungi

Each fungal strain was first cultivated on potato dextrose agar (PDA; HiMEDIA, India) at 25°C for seven days. Subsequently, three pieces (1 × 1-cm) of mycelium from these cultures were used to inoculate 100 mL of potato dextrose broth medium (potato infusion 200 g, dextrose 20 g, pH 5.1 ± 0.2) (PDB; HiMedia, India) in 250-mL Erlenmeyer flasks. Liquid cultures were grown for 7 days under static conditions in an incubator (Gallenkamp, UK) with a temperature set at 25 ± 2°C before extraction. After the incubation period, 100 mL of ethyl acetate (EtOAc) was added to each culture, gently shaken, and kept overnight at room temperature. The organic phase was then decanted, and the aqueous phase was extracted three times with 100 mL EtOAc. The organic phases were pooled (total = 300 mL) and evaporated to dryness at 40°C using a rotary evaporation system (Heidolph, Germany) to obtain crude fungal extracts. The dry residue was weighed (S1 Table in S1 File), diluted with 100% DMSO (Loba Chemie, India) to a stock concentration of 25 mg/mL, and stored in sterile conical flasks at 4°C until further use.

## Screening of fungal extracts for antibacterial activity

**Microbial species and culture media.** Three bacterial species were used in this investigation. Two strains, *Staphylococcus aureus* ATCC 43300 and *Escherichia coli* ATCC 25922, were obtained from the American Type Culture Collection (ATCC) and *Klebsiella oxytoca*, and a clinical isolate was obtained from the "Centre Pasteur du Cameroun" (CPC) of Yaoundé-Cameroon. Nutrient agar medium (NA; HiMedia, India) was used to revive the bacterial strains 24 h before each experiment, while nutrient broth (NB; HiMedia, India) medium was used for all antibacterial assays.

**Antibacterial screening and minimum inhibitory concentrations of endophytic extracts.** Individual ethyl acetate (EtOAc) extracts of endophytes were screened (at a single concentration) for antibacterial activity. Subsequently, the active compounds were selected for the dose–response study for MIC determination.

Extracts were screened against three bacterial species at a single concentration of 100 μg/mL according to the M07-A9 protocol of the Clinical Laboratory Standards Institute (CLSI) [22]. Briefly, 92 μL of nutrient broth (NB; HiMedia, India) was aseptically introduced into the wells of a 96-well microplate. Eight microliters (8 μL) of each extract, initially prepared in an intermediate plate at 2500 μg/mL, was added to the wells followed by 100 μL of standardized bacterial suspension (2 x10$^6$ CFU/mL). The tests were performed simultaneously for the negative control (NB + bacteria) and sterility control (NB alone). Ciprofloxacin (Sigma–Aldrich, Germany) at 64 μg/mL was used as the positive control. The test was performed in triplicate, and the plates were incubated at 37°C for 24 hours. After this period, extracts that were active

on at least one of the bacteria tested were selected to determine the minimum inhibitory concentrations (MICs).

To determine the MIC of the selected extracts, serial twofold dilutions of each extract were performed with final concentrations ranging from 1.5625 to 100 μg/mL for the extracts and from 0.117 to 7.5 μg/mL for the positive control. All test plates were incubated at 37°C for 24 hours. The turbidity was observed as an indication of growth, and the MIC was considered the lowest concentration with no visible growth of microorganisms (no turbidity). Wells containing NB and bacteria constituted the negative control, while the sterility control contained NB alone. The final concentration of DMSO (Sigma–Aldrich, Germany) was at most 1%, and the preliminary test did not inhibit bacterial growth. The test was performed in duplicate and repeated twice.

### Evaluation of possible modes of action of the most potent extracts

The five extracts (AMtw3, AMtf15, AMf6, AMf4, and AMsb23) exhibiting the best inhibitory activity against the three bacterial pathogens were selected for the mode of action study. The two most sensitive bacterial strains (*S. aureus* ATCC 43300 and *K. oxytoca*) were used for this study.

**Inhibition of Catalase Activity.** The catalase inhibitory activity of extracts was carried out as described by Mbekou *et al.* [23]. Extracts at the MIC concentration were added to a test tube containing 400 μL of hydrogen peroxide ($H_2O_2$) (40 mM) and 400 μL of phosphate buffer saline (PBS) (Sigma–Aldrich, Germany). The mixture was then transferred to another tube containing 200 μL of a bacterial suspension ($1.5 \times 10^8$ UFC/mL) prepared in 0.9% NaCl. The samples were incubated at 37°C for 30 min, after which they were centrifuged at 1200 rpm for 10 min. The supernatants were collected, and their optical density (OD) was read at 232 nm using the microplate reader Infinite M200 (TECAN, Männedorf, Switzerland). The phosphate buffer constituted the blank, while bacterial strains in phosphate buffer without any inhibiting substance were used as a negative control. The mixture of ciprofloxacin at 0.468 and 0.234 μg/mL, respectively, for *S. aureus* ATCC 43300 and *K. oxytoca*, phosphate buffer, and bacterial strain constituted a positive control. The percentage of remaining $H_2O_2$ was determined according to the following formula:

$$\% \text{ of remaining } H_2O_2 = \text{Abs sample} - \text{Abs negative control} \times 100/\text{Abs negative control}$$

Abs negative control is the absorbance of $H_2O_2$ without the extract, and Abs sample is the absorbance of $H_2O_2$ with the extract.

**Bacteriolysis activity of extracts.** The potential bacteriolytic ability of the antibacterial extracts was determined according to the method described by Mbekou *et al.* [23]. Briefly, the bacterial suspension ($5 \times 10^7$ CFU/mL) was treated with extracts at MIC, 2 MIC, and 4 MIC and incubated at 37°C. The optical density (OD) at 620 nm was measured at four different periods, including 0 h, 1 h, 2 h, and 4 h, using an Infinite M200 microplate reader (TECAN, Männedorf, Switzerland). A decrease in OD at 620 nm indicated bacterial cell lysis. Corresponding dilutions of the extract were used as blanks. Ciprofloxacin was used as a positive control at 0.468 and 0.234 μg/mL for *S. aureus* ATCC 43300 and *K. oxytoca*, respectively. The results were expressed as a ratio of the OD at each time interval versus the OD at 0 min (in %). All assays were carried out in triplicate.

**Outer membrane permeability assay.** The potential effect of extracts on the outer membrane (OM) permeability of *S. aureus* ATCC 43300 and *K. oxytoca* was determined according to the method described by Mbekou *et al.* [23]. An overnight culture ($5 \times 10^7$ CFU/mL) was inoculated into nutrient broth containing the extracts at 1/16 MIC, 1/8 MIC, 1/4 MIC, 1/2

MIC, MIC, 2 MIC, and 4 MIC. The media was then poured into sterilized 96-well microplates (100 μL) and incubated at 37°C for 24 h. After the incubation time, the growth of *S. aureus* ATCC 43300 and *K. oxytoca* was measured at 450 nm using an Infinite M200 microplate reader (TECAN, Männedorf, Switzerland). The bacterial growth graph (OD/450 nm) as a function of extract concentration (μg/mL) was plotted. Ciprofloxacin (concentration ranged from 0.01 to 0.93 μg/mL and 0.06 to 1.84 μg/mL, respectively, for *K. oxytoca* and *S. aureus* ATCC 43300) was used as the positive control, and each test was conducted in triplicate.

**Antibacterial time-kill kinetics assay.** The time-kill kinetics of active endophyte extracts were determined according to the method described by Babii *et al*. [24]. Here, extract concentrations of MIC, 2 MIC and 4 MIC were prepared by serial twofold dilution in a 96-well microplate. One hundred microliters (100 μL) of bacterial suspension (2 x 10$^6$ CFU/mL) was added, and the plate was incubated at 37°C for different time intervals (0, 1, 2, 4, 6, 8, 10, 12, and 24 h). Following each incubation period, the cell suspensions were appropriately diluted (in NaCl 0.9%), and the ODs of the resulting solution were measured at 620 nm using an Infinite M200 microplate reader (TECAN, Männedorf, Switzerland). Ciprofloxacin at 0.468 and 0.234 μg/ mL, respectively, for *S. aureus* ATCC 43300 and *K. oxytoca* was used as a positive control, while the bacteria incubated with NB were used as growth controls. The test was performed in triplicate, and the results are presented as the mean±SD.

**Biofilm inhibition assay.** *Biofilm Quantification*. Biofilm production by *S. aureus* ATCC 43300 and the *K. oxytoca* isolate was performed as described by Cruz *et al*. [25]. Briefly, a single colony was taken from the NA overnight bacterial culture, inoculated into 0.9% (w/v) saline solution and vortexed to ensure that the bacterial suspension was homogeneous. Bacterial suspensions were adjusted to $1 \times 10^7$ colony forming units (CFU/mL) by diluting with appropriate nutrient broth supplemented with 2% glucose. An aliquot of 200 μL of bacterial suspension per well was dispensed into a 96-well flat-bottom microplate. The plate was then incubated at 37°C for 48 hours. After this incubation period, planktonic cells were carefully removed, and adhered/biofilm cells were washed twice with 0.9% NaCl. Next, 100 μL of resazurin solution at 0.15 mg/mL prepared in sterile phosphate buffered saline (PBS; Sigma–Aldrich, Germany) was added to each well-containing biofilm. Microplates were incubated in the dark at 37°C for 1 hour, after which the microplate reader Infinite M200 (TECAN, Männedorf, Switzerland) was used to measure the relative fluorescence units (RFU) ($\lambda_{Ex}$ 530 nm and $\lambda_{Em}$ 590 nm). The relative fluorescence units obtained were used to plot a histogram to examine the amount of biofilms formed by each microorganism.

## Biofilm inhibition assay

The inhibitory potential of active extracts against biofilm formation by *S. aureus* ATCC 43300 and *K. oxytoca* was investigated using the method previously described by Cruz *et al*. [25] with slight modifications. Briefly, 100 μL of each bacterial strain suspension (1x10$^7$ CFU/mL) was incubated with crude extracts at MIC, 2 MIC, and 4 MIC for 48 h at 37°C. After incubation, planktonic cells were removed by washing the wells very delicately with 0.9% NaCl. Next, 100 μL of diluted resazurin solution was added to each well-containing biofilm. Microplates were placed in the dark and incubated at 37°C for 1 hour. A microplate reader Infinite M200 (TECAN, Männedorf, Switzerland) was then used to measure the relative fluorescence units (RFU) ($\lambda_{Ex}$ 530 nm and $\lambda_{Em}$ 590 nm) after incubation. Ciprofloxacin was used as a positive control at 0.468 and 0.234 μg/mL for *S. aureus* ATCC 43300 and *K. oxytoca*, respectively. Wells containing only bacteria-free medium constituted the negative control, and the assay was performed in triplicate and repeated twice. The percent inhibition was estimated (Biofilm inhibition% = RFU control − RFU sample/RFU control × 100), and the inhibitory

concentration 50 ($IC_{50s}$) was calculated based on the percent inhibition with the different concentrations of extracts.

## *In vitro* antioxidant activity of fungal extracts

All active endophytic fungal extracts were investigated for their potential antioxidant activity using DPPH radical scavenging and ferric ion reducing antioxidant power (FRAP) assays.

**DPPH radical scavenging assay.** The potential free radical scavenging activity of fungal extracts was evaluated using the 2,2-diphenyl-1-picryl-hydrazyl (DPPH) assay previously described by Djague *et al.*[26]. Briefly, the extracts were first diluted to obtain extract concentrations of 62.5, 125, 250, 500, 1000, 2000, and 4000 μg/mL in a 96-well microplate. After that, twenty-five microliters (25 μL) of each dilution was introduced into a new microplate, and 75 μL of 0.02% DPPH was added to obtain final concentrations of 15.625, 31.25, 62.5, 125, 250, 500, and 1000 μg/mL. The reaction mixtures were kept in the dark at room temperature for 30 min, after which the absorbance was measured at 517 nm against the blank (DPPH in methanol) using the microplate reader Infinite M200 (TECAN, Männedorf, Switzerland). L-ascorbic acid was used as a positive control and was treated under the same conditions as the extracts with final concentrations of 0.391, 0.781, 1.563, 3.125, 6.25, 12.5, and 25.0 μg/mL. The assay was performed in triplicate. The percentage (%) radical scavenging activities of the plant extracts were calculated using the following formula, from which other parameters, such as the radical scavenging activity 50 ($RSA_{50}$), the effective concentration 50 ($EC_{50}$), and the antiradical power (ARP), were deduced.

$$\text{Percentage of RSA} = [(\text{Abs control} - \text{Abs sample})/\text{Abs control}] \times 100.$$

where RSA is the radical scavenging activity, Abs control is the absorbance of the blank (DPPH + methanol), and Abs sample is the absorbance of DPPH radical + fungal extract.

**Ferric ion reducing antioxidant power (FRAP) assay.** The assay was performed according to the method described by Djague *et al.* [26]. Briefly, the extracts were first dissolved for the DPPH assay. Twenty-five microliters from each dilution was added to 25 μL of 1.2 mg/mL $Fe^{3+}$ solution in a new microplate. The plates were preincubated for 15 min at room temperature. Fifty microlitres (50 μL) of 0.2% ortho-phenanthroline solution was added to obtain final extract concentrations of 15.625, 31.25, 62.5, 125, 250, 500, and 1000 μg/mL. The reaction mixtures were further incubated for 20 min at room temperature, after which the absorbance was measured at 505 nm using a 96-well microplate reader Infinite M200 (TECAN, Männedorf, Switzerland) against the blank (25 μL methanol + 25 μL $Fe^{3+}$ + 50 μL ortho-phenanthroline). Hydroxylamine was used as a positive control and was treated in the same way as the extracts with final concentrations of 0.103, 0.206, 0.413, 0.825, 1.65, 3.30, and 6.60 μg/mL. The assay was performed in triplicate. From a concentration-activity curve of $NH_2OH$ used as a standard, the optical densities of the test wells were projected, and the results were expressed quantitatively as μg equivalent $NH_2OH$/g of extracts.

## Cytotoxicity assay of promising extracts

The cytotoxic effect of antibacterial extracts was assessed using the MTT assay as described by Mosmann, [27], targeting normal monkey kidney Vero cells ATCC CRL1586 cultured in complete medium containing 13.5 g/L DMEM (Gibco, Waltham, MA USA), 10% fetal bovine serum (Gibco, Waltham, MA, USA), 0.21% sodium bicarbonate (Sigma–Aldrich, New Delhi, India), and 50 μg/mL gentamicin (Gibco, Waltham, MA, USA). Essentially, Vero cells at $5 \times 10^3$ cells/200 μL/well were seeded into 96-well flat-bottomed tissue culture plates (Corning, USA) in complete medium. Fifty microliters of serially diluted extract solutions (≤100 μg/mL) were

added after 24 h of seeding, and the cells plus test substance were incubated for 48 h in a humidified atmosphere at 37°C and 5% $CO_2$. DMSO (0.4% v/v) was added as a negative control (100% growth). Twenty microliters of a stock solution of MTT (5 mg/mL in 1x phosphate-buffered saline) was added to each well, gently mixed, and incubated for an additional 4 h. After spinning the plate at 1500 rpm for 5 min, the supernatant was carefully removed, and 100 μL of 100% DMSO (v/v) was added to dissolve the formazan. The plate was read on a microtiter plate reader Infinite M200 (TECAN, Männedorf, Switzerland) at 570 nm. The 50% cytotoxic concentrations ($CC_{50}$) of extracts were determined by analyzing the dose–response curves.

## Statistical analysis

Data collected from at least three independent experiments were analyzed using one-way ANOVA with GraphPad Prism 5. Data are expressed as the mean ± SD of experiments performed in triplicate. Error bars represent the SD, and significant differences for multiple comparisons were determined by the Turkey test at $p < 0.05$.

## Results

### Screening and characterization of potent endophytic fungal strains

**Antibacterial screening.** In the present study, forty-one endophytic fungi isolated from ten organs of the medicinal plant *A. muricata* growing in Cameroon were subjected to antibacterial screening against three pathogenic bacteria causing UTIs. Fungi fermented in potato dextrose broth medium and extracted with ethyl acetate yielded crude extracts with masses ranging from 12 to 402 mg depending on the cultured fungus. AMs1 (402 mg) followed by AMs3 (264 mg) and AMrb9 (258 mg) isolated from seeds (s) and root bark (rb) produced the highest amount of crude metabolites (S1 Table in S1 File).

The 41 crude extracts were screened at a single concentration of 100 μg/mL against three bacterial causative agents of UTIs (*E. coli* ATCC 25922, *K. oxytoca*, and *S. aureus* ATCC 43300). The results (S1 Table in S1 File) showed that 17 (41.46%) were active, among which 16 exhibited broad-spectrum activity against the three pathogens, while one (AMr10) was only active against *S. aureus* ATCC 43300. Crude metabolites exhibiting activity were produced by endophytic fungi isolated from root (2), root bark (3), fruits (4), seeds (2), the thorn of fruits (2), twigs (1), bark (1), and stem bark (2). No endophytes from leaves or peduncles showed any inhibitory activity against the tested pathogens. Based on their antibacterial activity at 100 μg/mL, the seventeen fungi were selected for molecular identification and further studies.

**Molecular characterization of endophytic fungi.** The ITS rDNA region of all 17 selected fungi was sequenced, and the identification was performed by comparison with published sequences in GenBank (S2 File in S1 File). The results from the BLAST search revealed that 16 endophyte sequences showed 98.32–100% similarity with sequences from previously identified fungi in the NCBI database. However, eleven of the endophytes sequences investigated showed higher similarity with more than two different fungal species. AMr9 showed similarity with six different species of *Aspergillus*, and AMtf15, AMr10, AMrb9, AMs9, and AMb7 were similar to 3 different *Aspergillus* spp. AMrb1 and AMrb11 were similar to three *Penicillium* species, AMf4 was similar to two *Curvularia* species, and AMf3 was similar to more than two *Meyerozyma* species. The sequence of AMtf5 showed only 96.88% similarity with the strain *Talaromyces clemensii* NR_168822.1. New sequences generated in this study were deposited in the NCBI GenBank nucleotide database (www.ncbi.nlm.nih.gov; Table 1).

To infer the evolutionary history of endophytic fungi, the maximum likelihood (ML) method based on the Kimura 2-parameter model was used (Fig 1). The percentage of replicate trees in which the associated taxa clustered together in the bootstrap test (1000 replicates) are

**Table 1. Identity of the seventeen active endophytic fungi, their GenBank accession number, and their closest related species.**

| Organs | Fungus code | Fungal identity | GenBank Accession No. | Blast Results | | |
|---|---|---|---|---|---|---|
| | | | | Closest Related Species | Query coverage (%) | Percent identity (%) |
| **Root** | AMr9 | *Aspergillus* sp. | OM959526 | *A. foetidus* NR_163668.1 | 100 | 100 |
| | | | | *A. welwitschiae* NR_137513.1 | 100 | 100 |
| | | | | *A. awamori* NR_077143.1 | 100 | 100 |
| | | | | *A. tubingensis* NR_131293.1 | 100 | 99.60 |
| | | | | *A. costaricensis* NR_103604.1 | 100 | 99.60 |
| | | | | *A. niger* NR_111348.1 | 98 | 100 |
| | AMr10 | *Aspergillus* sp. | OM959517 | *A. austwickii* NR_171607.1 | 100 | 99.60 |
| | | | | *A. aflatoxiformans* NR_171606.1 | 100 | 99.60 |
| | | | | *A. oryzae* NR_135395.1 | 100 | 99.40 |
| **Root bark** | AMrb1 | *Penicillium* sp. | OM959527 | *P. herquei* NR_103659.1 | 100 | 99.26 |
| | | | | *P. verrucisporum* NR_153275.1 | 100 | 99.26 |
| | | | | *P. sanshaense* NR_153276.1 | 100 | 98.76 |
| | AMrb9 | *Aspergillus* sp. | OM959523 | *A. clavatonanicus* NR_135410.1 | 100 | 100 |
| | | | | *A. longivesica* NR_135411.1 | 100 | 99.80 |
| | | | | *A. giganteus* NR_135403.1 | 100 | 99.80 |
| | AMrb11 | *Penicillium* sp. | OM959528 | *P. citrinum* NR_121224.1 | 100 | 99.40 |
| **Fruit** | AMf6 | *Penicillium citrinum* | OM980761 | *P. citrinum* NR_121224.1 | 100 | 99.53 |
| | AMf3 | *Meyerozyma* sp. | OM959515 | *M. caribbica* NR_149348.1 | 99 | 99.57 |
| | | | | *M. carpophila* NR_152984.1 | 100 | 100 |
| | AMf4 | *Curvularia* sp. | OM959516 | *C. kenpeggii* NR_158447.1 | 100 | 98.32 |
| | AMf1 | *Penicillium citrinum* | OM959525 | *P.citrinum* NR_121224.1 | 100 | 99.56 |
| **Seed** | AMs3 | *Penicillium citrinum* | OM959520 | *P. citrinum* NR_121224.1 | 100 | 98.68 |
| | AMs9 | *Aspergillus* sp. | OM959518 | *A. clavatonanicus* NR_135410.1 | 100 | 100 |
| | | | | *A. longivesica* NR_135411.1 | 100 | 99.79 |
| | | | | *A. giganteus* NR_135403.1 | 100 | 99.79 |
| **Thorn of fruit** | AMtf15 | *Aspergillus* sp. | OM959513 | *A. austwickii* NR_171607.1 | 100 | 99.57% |
| | AMtf5 | *Talaromyces* sp. | OM959521 | *T. clemensii* NR_168822.1 | 100 | 96.88 |
| **Twig** | AMtw3 | *Fusarium waltergamsii* | OM959524 | *F. waltergamsii* NR_159548.1 | 93 | 99.77 |
| **Stem bark** | AMsb23 | *Talaromyces annesophieae* | OM959519 | *T. annesophieae* NR_170732.1 | 100 | 99.17 |
| | AMsb1 | *Penicillium citrinum* | OM959514 | *P. citrinum* NR_121224.1 | 100 | 99.53 |
| **Bark** | AMb7 | *Aspergillus* sp. | OM959522 | *A. austwickii* NR_171607.1 | 100 | 99.60 |
| | | | | *A. flatoxiformans* NR_171606.1 | 100 | 99.60 |
| | | | | *A. oryzae* NR_135395.1 | 100 | 99.40 |

AMr: *A. muricata* root, AMrb: *A. muricata* root bark, AMf: *A. muricata* fruit, AMs: *A. muricata* seed, AMtf: *A. muricata* thorn of fruit, AMtw: *A. muricata* twigs, AMl: *A. muricata* leaves, AMb: *A. muricata* bark, AMsb: *A. muricata* stem bark, AMpe: *A. muricata* peducle.

shown next to the branches. The tree is drawn to scale, with branch lengths in the same units as the evolutionary distances used to infer the phylogenetic tree. With the bootstrap support of each clade over 80, the ML phylogenetic analysis showed that the seventeen endophytic fungi belonged to six genera from the phylum Ascomycota, including *Aspergillus*, *Curvularia*, *Fusarium*, *Meyerozyma*, *Penicillium*, and *Talaromyces*. Confirming the results from the BLAST search, eleven endophytes (AMf15, AMr10, AMb7, AMs9, AMrb9, AMrb11, AMf4, AMf3, AMtf5, AMrb1, and AMr9) were classified at the genus level. In comparison, only six endophytes could be identified at the species level, including AMf6, AMsb1, AMs3 and AMf1 as *Penicillium citrinum*, AMsb23 as *Talaromyces annesophieae* and AMtw3 as *Fusarium*

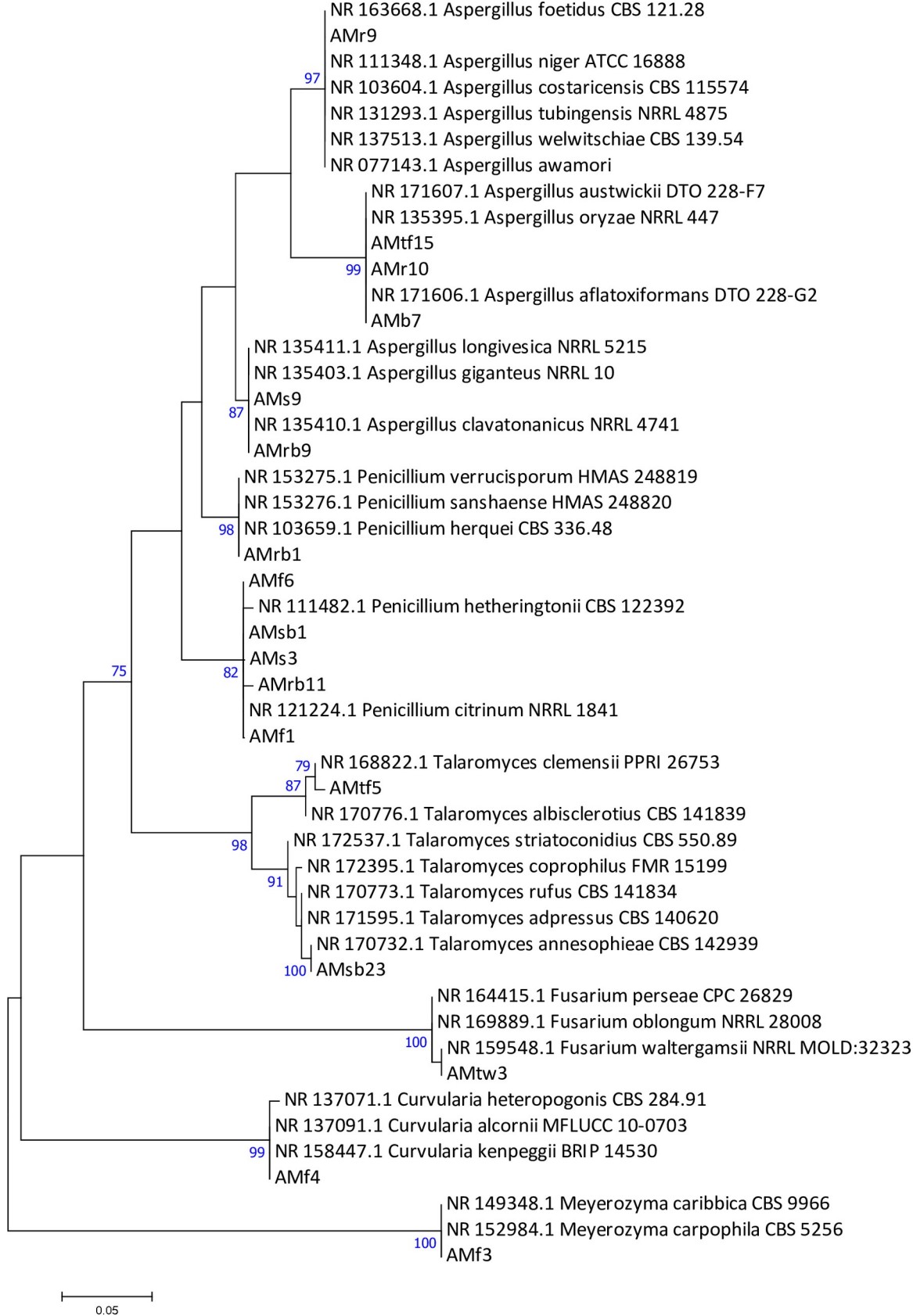

**Fig 1. Molecular phylogenetic tree generated by Maximum Likelihood analysis based on ITS sequence alignments of endophytic fungal isolates.** ML bootstrap support values (ML $\geq$ 70%) are shown at the nodes. Isolates from *Annona muricata* are coded. The scale bar indicates 0.05 expected changes per site. The evolutionary history was inferred by using the Maximum Likelihood method based on the Kimura 2-parameter model. The tree with the highest log likelihood (-1028.13) is shown. The percentage of trees in which the associated taxa clustered together is shown next to the branches. Initial tree(s) for the heuristic search were obtained automatically by applying the Maximum Parsimony method. A discrete Gamma distribution was used to model evolutionary rate differences among sites (2 categories (+$G$, parameter = 0.4638)). The tree is drawn to scale, with branch lengths measured in the number of substitutions per site. The analysis involved 49 nucleotide sequences. Codon positions included were 1st+2nd+3rd+Noncoding. All positions containing gaps and missing data were eliminated. There were a total of 221 positions in the final dataset. Evolutionary analyses were conducted in MEGA7.

*waltergamsii*. Overall, fungal species from the *Aspergillus* genus were identified in five organs: root (2), bark, seed, root bark, and thorn of fruits. *Penicillium* species were identified among fungi isolated from fruits (2), root bark (2), stem bark, and seeds. *Talaromyces* species were identified in the thorns of fruits and stembarks, while *Myerozyma*, *Fusarium*, and *Curvularia* were identified only in fruits, twigs, and fruits, respectively. Overall, the 17 selected fungi were mainly dominated by *Penicillium* and *Aspergillus*, accounting for 35.29% of the total.

## Minimum inhibition concentrations (MICs) and cytotoxic concentrations 50 (CC$_{50}$) of selected extracts

The 17 selected crude extracts were submitted to a dose–response study for MIC determination, and the results are summarized in Table 2. All extracts exhibited activity against at least one of the tested bacteria, with MIC values ranging from 3.125 to 100 µg/mL depending on the extracts and microorganisms. Extract from *F. waltergamsii* AMtw3 (3.125 µg/mL) was the most active against *Staphylococcus aureus* ATCC 43300, followed by *Aspergillus* sp. AMtf15, *P. citrinum* AMf6 and *P. citrinum* AMs3 (MIC 9.375 µg/mL). When tested against *K. oxytoca*, extracts *F. waltergamsii* AMtw3 (MIC 3.125 µg/mL) and *P. citrinum* AMsb1 (MIC 3.125 µg/mL) were the most active, followed by *P. citrinum* AMf6 (MIC 4.687 µg/mL). Against *E. coli* ATCC 25922, *F. waltergamsii* AMtw3 (MIC 6.25 µg/mL) exhibited the best activity, followed by *P. citrinum* AMf6 and *P. citrinum* AMsb1 (MIC 9.375 µg/mL).

The results from the dose–response study also revealed that extracts produced by endophytes from the same genus and isolated from the same or different organs displayed very different potency and activity profiles. For instance, *Aspergillus* sp. AMr10 was inactive (MIC > 100 µg/mL) against *K. oxytoca* and *E. coli* ATCC 25922 and displayed weak activity against *S. aureus* ATCC 43300 (MIC 100 µg/mL), while *Aspergillus* sp. AMr9 isolated from the same organ (root) was very active against the three bacteria (MIC 12.5 µg/mL). The extract from *P. citrinum* AMsb1 from stem bark was four times more potent against *K. oxytoca* (MIC 3.125 µg/mL) than the extract produced by *P. citrinum* AMs3 isolated from seeds (MIC 12.5 µg/mL). Conversely, *Aspergillus* spp. AMr9, AMrb9, and AMs9 isolated from roots, root bark, and seeds displayed a similar activity profile against the three pathogens (Table 2). In addition to their antibacterial activity, all selected extracts displayed weak cytotoxicity against Vero cells ATCC CRL1586 with a median cytotoxic concentration (CC$_{50}$) greater than 100 µg/mL. Extracts from *F. waltergamsii* AMtw3, *P. citrinum* AMf6, *Aspergillus* sp. AMtf15, *Curvularia* sp. AMf4 and *T. annesophieae* AMsb23, produced by fungi from different genera, were selected to investigate their potential antibacterial mode of action.

## Modes of action of promising extracts

**Effect of extracts on catalase activity.** The inhibition of the catalase activity of *S. aureus* ATCC 43300 and *K. oxytoca* was evaluated by comparing the amount of H$_2$O$_2$ remaining

**Table 2. Yields (mg), minimum inhibitory concentrations (MICs) and cytotoxic concentrations 50 ($CC_{50}$) of selected extracts.**

| Plant organs | Fungal Extracts | Yield (mg) | MIC (µg/mL ± SD) | | | $CC_{50}$ (µg/mL ± SD) |
|---|---|---|---|---|---|---|
| | | | *S. aureus* ATCC 43300 | *K. oxytoca* | *E. coli* ATCC 25922 | Vero cells ATCC CRL1586 |
| **Roots** | *Aspergillus* sp. AMr9 | 195 | 12.5±0.00 | 12.5±0.00 | 12.5±0.00 | > 100 |
| | *Aspergillus* sp. AMr10 | 12 | 100±0.00 | > 100 | > 100 | > 100 |
| **Root bark** | *Penicillium* sp. AMrb1 | 95 | 18.75±8.83 | 12.5±0.00 | 12.5±0.00 | > 100 |
| | *Aspergillus* sp. AMrb9 | 258 | 12.5±0.00 | 12.5±0.00 | 12.5±0.00 | > 100 |
| | *Penicillium* sp. AMrb11 | 61 | 12.5±0.00 | 12.5±0.00 | 12.5±0.00 | > 100 |
| **Fruits** | *Penicillium citrinum* AMf6 | 22 | 9.375±4.19 | 4.687±2.20 | 9.375±4.41 | > 100 |
| | *Meyerozyma* sp. AMf3 | 67 | 50±0.00 | 25±0.00 | 37.5±17.67 | > 100 |
| | *Curvularia* sp. AMf4 | 216 | 12.5±0.00 | 9.375±4.41 | 12.5±0.00 | > 100 |
| | *Penicillium citrinum* AMf1 | 126 | 12.5±0.00 | 9.375±4.41 | 12.5±0.00 | > 100 |
| **Seeds** | *Penicillium citrinum* AMs3 | 264 | 9.375±4.19 | 12.5±0.00 | 25±0.00 | > 100 |
| | *Aspergillus* sp. AMs9 | 139 | 12.5±0.00 | 12.5±0.00 | 18.75±8.83 | > 100 |
| **Thorns of fruit** | *Aspergillus* sp. AMtf15 | 56 | 9.375±4.41 | 6.25±0.00 | 12.5±0.00 | > 100 |
| | *Talaromyces* sp. AMtf5 | 95 | 18.75±8.83 | 6.25±0.00 | 12.5±0.00 | > 100 |
| **Twigs** | *Fusarium waltergamsii* AMtw3 | 48 | 3.125±0.00 | 3.125±0.00 | 6.25±0.00 | > 100 |
| **Stem bark** | *Talaromyces annesophieae* AMsb23 | 25 | 12.5±0.00 | 6.25±0.00 | 18.75±8.83 | > 100 |
| | *Penicillium citrinum* AMsb1 | 194 | 12.5±0.00 | 3.125±0.00 | 9.375±4.41 | > 100 |
| **Barks** | *Aspergillus* sp. AMb7 | 24 | 12.5±0.00 | 6.25±0.00 | 18.75±8.83 | > 100 |
| **Ciprofloxacin** | | NA | 0.468±0.00 | 0.234±0.00 | 0.234±0.00 | NA |
| **Podophyllotoxin** | | NA | NA | NA | NA | 0.177±0.05 |

AMr: *A. muricata* root; AMrb: *A. muricata* root bark; AMf: *A. muricata* fruit; AMs: *A. muricata* seed; AMtf: *A. muricata* thorn of fruit; AMtw: *A. muricata* twigs; AMb: *A. muricata* bark; AMsb: *A. muricata* stem bark; $CC_{50}$: Cytotoxic concentrations 50; MIC: Minimum inhibitory concentration; NA: Not applicable.

in the medium after the addition of fungal extracts to the control (Table 3). The percentages of remaining $H_2O_2$ in bacterial cultures treated with extracts ranged from 49.17–54.58% and 46.99–55.70% for *S. aureus* ATCC 43300 and *K. oxytoca*, respectively, highlighting the ability of the tested extracts to exert a certain degree of inhibition against the activity of the bacterial catalase enzyme. The catalase inhibition activity exhibited by all fungal extracts was comparable to that of the positive control, ciprofloxacin, since no statistically significant difference was observed ($p > 0.05$).

**Table 3. Percentage of remaining $H_2O_2$ after evaluation of the effect of endophytic fungal extracts on catalase activity of *S. aureus* ATCC 43300 and *K. oxytoca* at the MIC concentration.**

| Fungal extracts | Remaining $H_2O_2$ (%) ± SD | |
|---|---|---|
| | *K. oxytoca* | *S. aureus* ATCC 43300 |
| **F.waltergamsii AMtw3** | 52.20±1.67[ab] | 51.66±1.50[a] |
| **Aspergillus sp. AMtf15** | 55.70±0.00[a] | 51.84±3.77[a] |
| **P. citrinum AMf6** | 51.61±3.15[ab] | 54.58±2.36[a] |
| **Curvularia sp AMf4** | 53.68±0.95[ab] | 50.23±4.31[a] |
| **T.annesophieae AMsb23** | 46.99±1.78[b] | 49.17±3.53[a] |
| **Ciprofloxacin** | 54.12±2.89[ab] | 53.59±2.70[a] |

Data are expressed as the mean±SD. Along the column, values carrying the same letter superscripts are not significantly different ($p > 0.05$), and values carrying different letters are significantly different ($P < 0.05$). AMf: *A. muricata* fruit, AMtf: *A. muricata* thorn of fruit, AMtw: *A. muricata* twigs, AMsb: *A. muricata* stem bark.

## Lytic effect of extracts

The bacteriolysis assay was performed to investigate if active extracts inhibit *S. aureus* ATCC 43300 and *K. oxytoca* through cell lysis. Because viable bacteria absorb light at 620 nm, any decrease in optical density at this wavelength could be used as an indication of bacteriolysis. The cell lysis activity exhibited by fungal extracts is summarized in Fig 2. Globally, the treatment of the bacterial cells with fungal extracts (AMtw3, AMtf15, AMf6, AMf4, and AMsb23) caused cell lysis at all tested concentrations significantly after 4 h of incubation. Extracts from *F. waltergamsii* AMtw3 (70.95%, 62.25%) and *Aspergillus* sp. AMtf15 (61.9%, 54.01%) exhibited the highest bacteriolytic activity at 4 MIC against *K. oxytoca* and *S. aureus*. The reduction in the bacterial population was more pronounced against *K. oxytoca*, with relative absorbance percentages varying from 37.75–100%.

**Effect of extracts on the permeability of the outer cell membrane.** The capacity of endophytic extracts to permeabilise the outer cell membrane was determined throughout 24 h at seven different concentrations. Fig 3 shows that the permeabilisation activity of extracts was concentration dependent. Overall, all extracts resulted in a decrease in the optical density with increasing concentrations of extracts. This trend, therefore, could indicate the loss of intracellular ingredients by the chelation of divalent cations. It can also be noted that the destabilization power of the outer membrane of both bacteria by the different extracts was generally not significantly different ($p>0.05$) from that of ciprofloxacin at their corresponding concentrations.

**Bacterial growth curves.** The growth kinetics of bacterial strains exposed to three different concentrations (MIC, 2MIC and 4MIC values) of extracts for 24 h were compared to bacterial growth without antibiotic treatment (Fig 4). As revealed by the optical densities at 620 nm, all extracts at all tested concentrations significantly inhibited the growth of the two bacterial pathogens compared to the negative control. Globally, although there was an initial increase in the number of viable cells during the first 2–3 hours of incubation, a gradual decrease in the bacterial population was noted from the 3rd to the 24th hour depending on the extracts and the microorganism (Fig 4). The maximum bactericidal effect was noted after 24 h of incubation, with a decrease in the bacterial population by 1.78–7.42 times for *S. aureus* ATCC 43300 and 1.80–7.76 times for *K. oxytoca*. The extract from *F. waltergamsii* AMtw3 was the most efficient in killing *K. oxytoca* at all tested concentrations, with 5.80-, 6.14- and 7.76-fold reductions in the bacterial population at MIC, 2 MIC and 4 MIC, respectively. Against *S. aureus* ATCC 43300, the extract from *T. annesophieae* AMsb23 with 4.49-, 4.56-, and 7.42-fold reductions in the bacterial population at MIC, 2 MIC, and 4 MIC, respectively, was the most active. The activity of the most active extracts against both pathogens was significantly higher ($p\leq0.05$) than that of the positive control (ciprofloxacin) at MIC, which is known to exert a bactericidal effect (S3 Table in S1 File).

**Effect of endophytic extracts on biofilm formation.** *Biofilm quantification.* Biofilm production by *K. oxytoca* and *S. aureus* ATCC 43300 in NB medium supplemented with 2% glucose after 48 hours of incubation was expressed in terms of the mean fluorescence values from independent assays performed in triplicate. Fig 5 shows that after 48 hours of incubation, both *K. oxytoca* and *S. aureus* ATCC 43300 could form a significant amount of biofilm materialised by the mean relative fluorescence units (RFU) ($\lambda_{Ex}$ 530 nm and $\lambda_{Em}$ 590 nm) from two independent assays over 10000. There was no significant difference ($P\leq0.05$) between the amounts of biofilm formed by the two bacteria.

## Biofilm formation inhibition

Bacterial biofilms play a significantly important role in urinary tract infections (UTIs), which are responsible for persistent infections causing relapses and acute prostatitis [28]. Bacterial

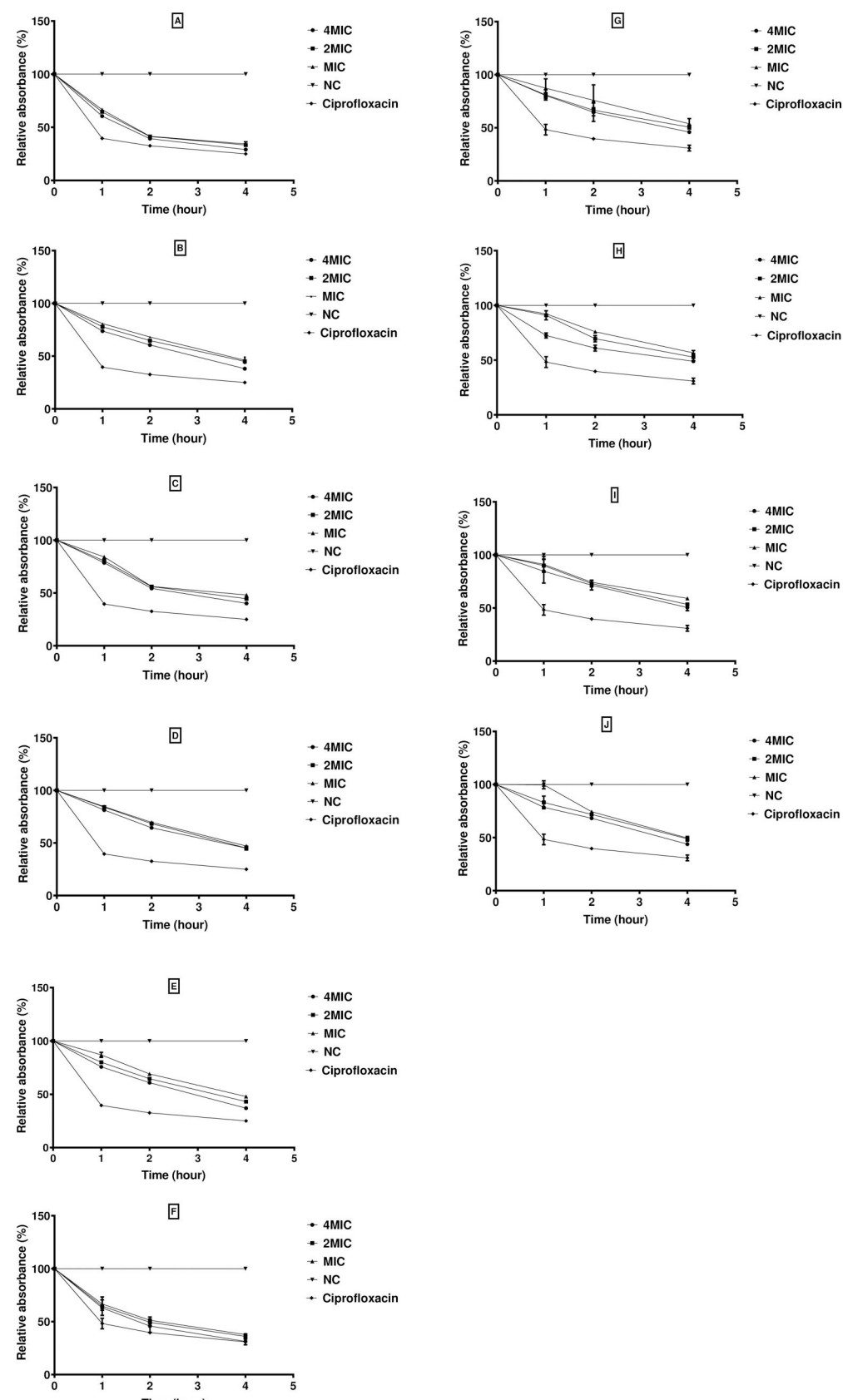

**Fig 2. The bacteriolytic activity of endophytic fungi extracts on *S. aureus* ATCC 43300, and *K. oxytoca*.** (A) AMtw3 on *K.oxytoca*; B) AMtf15 on *K. oxytoca*; (C) AMf6 on *K. oxytoca*; (D) AMf4 on *K. oxytoca*; (E) AMsb23 on *K. oxytoca*; (F) AMtw3 on *S. aureus*; (G) AMtf15 on *S. aureus*, (H) AMf6 on *S. aureus*; (I) AMf4 on *S. aureus*; (J) AMsb23 on *S. aureus*. AMf: *A. muricata* fruit, AMtf: *A. muricata* thorn of fruit, AMtw: *A. muricata* twigs, AMsb: *A. muricata* stem bark, MIC: Minimal Inhibitory Concentration, NC: Negative Control. Data are expressed as the mean±SD.

biofilm formation is difficult to eradicate due to several factors, including persistent cells showing reduced metabolism that leads to higher levels of antimicrobial resistance and the natural resistance conferred to bacteria by the biofilm structure [28]. Therefore, the search for new therapeutic agents is necessary. In this respect, the selected active extracts were also evaluated for their potential to inhibit biofilm formation by *S. aureus* ATCC 43300 and *K. oxytoca*. Determining the inhibitory concentration 50 ($IC_{50}$), defined as the concentration of extract required to reduce biofilm formation by half, revealed that extracts exhibited different degrees of inhibition against both bacterial pathogens (Table 4). The $IC_{50}$ values ranged from 0.25–11.86 µg/mL against *K. oxytoca* and 0.36–11.08 µg/mL depending on the extract and the microorganism. The extract from *F. waltergamsii* AMtw3 was consistently the most potent against the two bacteria ($IC_{50}$ 0.24 and 0.36 µg/mL), followed by *Aspergillus* sp. AMtf15 ($IC_{50}$, 0.73 and 1.903 µg/mL). *K. oxytoca* was more sensitive than *S. aureus*. Extracts from AMf6, AMtf15, AMtw3, and AMsb23 were 1.81, 2.60, 1.44 and 1.27 times more potent against biofilm formation by *K. oxytoca* than *S. aureus*.

## Antioxidant activity of fungal extracts

**DPPH radical scavenging activity.** The ability of the 17 antibacterial extracts to scavenge the free radical DPPH was measured at an absorbance of 517 nm. From the results, the radical scavenging activity of the tested extracts ranged from 146.05 to > 1000 µg/mL. Globally, among the 17 extracts tested, only 7 showed radical scavenging activity 50 ($RSA_{50}$) at less than 1000 µg/mL (Table 5). Although their activity was significantly lower ($p \leq 0.05$) than that of ascorbic acid used as a standard ($RSA_{50}$ 8.92 µg/mL), the extract from *T. clemensii* AMtf5 exhibited the highest scavenging activity ($RSA_{50}$ 146.05 µg/mL), followed by an extract from *Aspergillus* sp. AMr9 ($RSA_{50}$ 176.9 µg/mL) obtained from the thorns of fruits and roots of *A. muricata*, respectively. Interestingly, although *Aspergillus* spp. from roots displayed good potency, another *Aspergillus* isolate from the thorn of fruits was weakly active ($RSA_{50}$ 663.35 µg/mL). Similarly, while the extract from *P. citrinum* AMsb1 from stem bark was potent ($RSA_{50}$ 211.30 µg/mL), the *P. citrinum* AMf6 from fruits was the less active ($RSA_{50}$ 799.75 µg/mL) of the tested extracts. The same difference in activity could be observed with the *Talaromyces* species from the thorn of fruits ($RSA_{50}$ 146.05 µg/mL) and stem bark ($RSA_{50}$ 536.15 µg/mL). These observations could suggest that the organ of isolation of endophytic isolates could substantially influence the potency of each microbial species.

**Reduction of $Fe^{3+}$ ions by ortho-phenanthroline.** The seventeen extracts were also screened for their ability to reduce $Fe^{3+}$ to $Fe^{2+}$. The results from Table 6 show a significant correlation between the concentration of the extracts and their reducing power. By projecting the optical densities of the extracts on a concentration-activity curve of $NH_2OH$ used as standard, the results showed that only extracts from five endophytic fungi, including AMtf15, AMf6, AMs3, AMf4 and AMf1, exhibited activity (12.03, 12.28, 11.97, 12.35 and 12.37 µg equivalent $NH_2OH$/g, respectively) at 1000 µg/mL. Moreover, at concentrations of 1000, 500, 250 and 125 µg/mL, the ferric ion reducing capacities of all five extracts were not significantly different (P>0.05).

Out of the seventeen extracts screened for antioxidant activity, extracts from *Aspergillus* sp. AMtf15 and *P. citrinum* AMf6 exhibited DPPH radical scavenging and ferric ion reducing

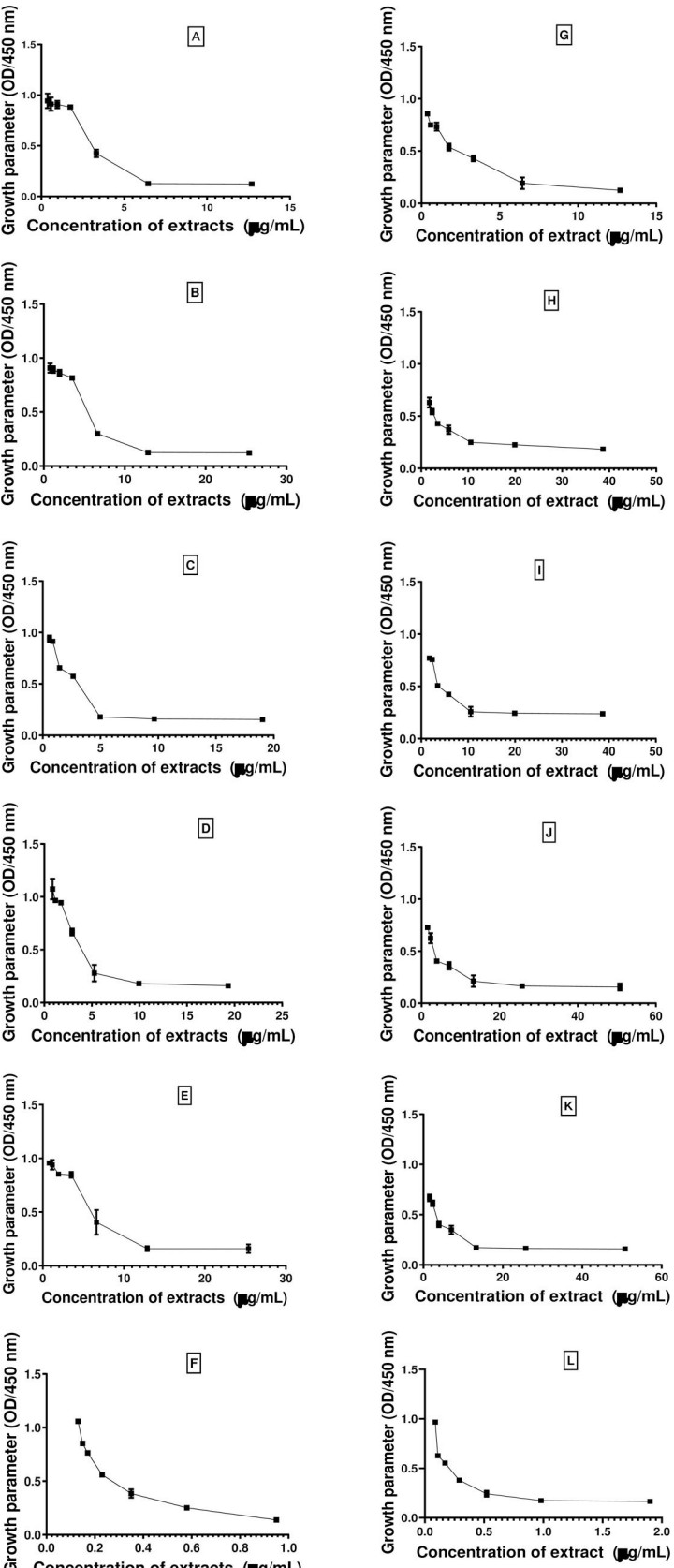

**Fig 3. Effect of endophytic fungi extracts on the outer membrane permeability of *S. aureus* ATCC 43300 and *K. oxytoca*.** (A) AMtw3 on *K. oxytoca*. B) AMtf15 on *K. oxytoca*; (C) AMf6 on *K. oxytoca*; (D) AMf4 on *K. oxytoca*; (E) AMsb23 on *K. oxytoca*; (F) Ciprofoxacin on *K. oxytoca*; (G) AMtw3 on *S. aureus*; (H) AMtf15 on *Staphylococcus aureus*; (I) AMf6 on *S. aureus*; (J) AMf4 on *S. aureus*; (K) AMsb23 on *S. aureus*; (L) Ciprofloxacin on *S. aureus*. AMf: *A. muricata* fruit, AMtf: *A. muricata* thorn of fruit, AMtw: *A. muricata* twigs, AMsb: *A. muricata* stem bark. Data are expressed as the mean±SD.

capacities. However, extracts from AMtf5, AMtw3, AMr9, AMsb23 and AMsb1 displayed DPPH radical scavenging activity and very weak ferric ion reducing capacities, while AMs3, AMf4 and AMf1 only showed ferric ion reducing capacities.

## Discussion

Antimicrobial resistance is one of the most critical global public health threats of the 21st century. Faced with this reality, the need for action, particularly the search for and development of new antibiotics to avert a developing global health care crisis, is imperative [29]. Natural products, mainly from microbial origins with their widely divergent chemical structures, have been a prolific source and an inspiration for numerous antibiotic agents [30]. Over the last decades, endophytic fungi inhabiting various medicinal plants have gained tremendous attention due to their ability to produce novel bioactive compounds exhibiting various biological properties, including antibacterial properties [12, 15, 16]. Moreover, antibacterial compounds produced by endophytes have been shown to occupy a broad spectrum of structural classes, such as alkaloids, peptides, steroids, terpenoids, phenols, quinines, and flavonoids [31, 32]. The extensive tropical rainforests of Cameroon rich with thousands of medicinal plants can be an excellent bioresource for plant-associated microorganisms with the potential to produce novel and highly potent antimicrobial compounds [18, 23]. Therefore, the current study was designed to investigate the antibacterial and antioxidant potential of extracts from endophytic fungi associated with *Annona muricata*, a plant traditionally used in Cameroon to treat microbial infections.

Crude ethyl acetate extracts from forty-one (41) endophytic fungi isolated from distinct organs of *A. muricata* were screened for their ability to inhibit *Escherichia coli* ATCC 25922, *Klebsiella oxytoca*, and *Staphylococcus aureus* ATCC 43300. Of the 41 extracts tested, 17 (41.46%) exhibited activity against at least one bacteria. Our previous screening of 56 extracts from endophytic fungi isolated from the three Cameroonian medicinal plants, *Terminalia mantaly*, *Terminalia catappa*, and *Cananga odorata*, against seven bacterial strains revealed that approximately 13% were very active against all tested bacterial strains [23]. Another previous screening of 152 extracts from endophytic fungi from *A. muricata* against *Plasmodium* parasites revealed that over 17% of isolates exhibited activity [18]. These results indicate that many endophytes inhabiting Cameroonian plants could produce active metabolites to inhibit various human pathogenic microbes. The seventeen active endophytes identified by sequencing their ITS1-5.8S rRNA-ITS2 region belonged to the *Aspergillus*, *Curvularia*, *Fusarium*, *Meyerozyma*, *Penicillium*, and *Talaromyces* genera. Among them, 11 (64.70%) were identified at the genus level, while only six (35.29%) were identified at the species level. Given the inability of the ITS sequence to give precise identification of our endophytic isolates, further analysis, such as the sequencing of additional genes followed by multilocus phylogenetic analysis, will be conducted to achieve the precise identification of these isolates [33–35]. However, the great diversity of fungal genera identified from the present study indicates that *A. muricata* hosts a large group of microorganisms capable of producing antibacterial metabolites. The extracts from the seventeen identified isolates were submitted to a dose–response study for MIC determination. All extracts exhibited activity against all tested bacteria, with MIC values ranging

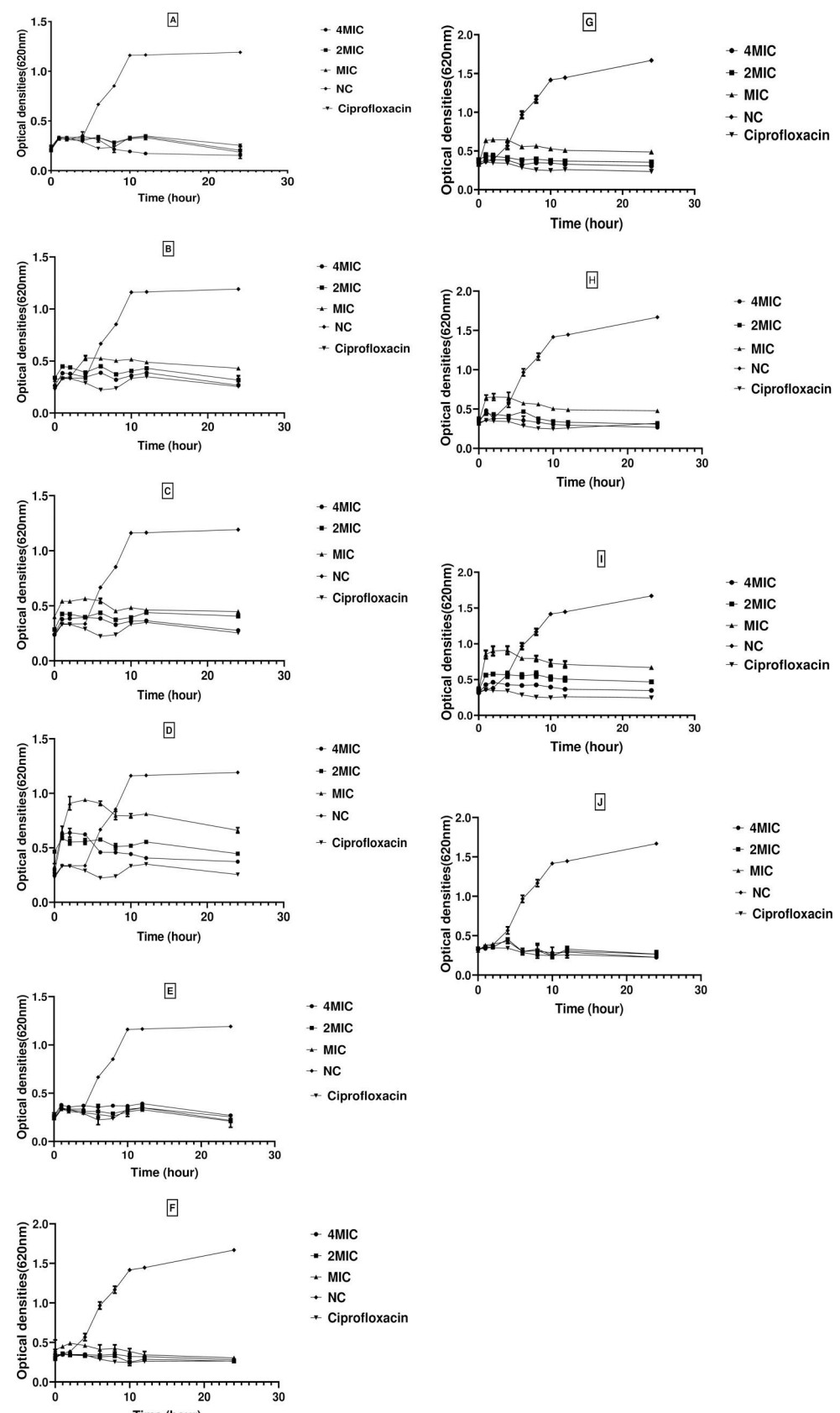

**Fig 4. Activity kinetics of extracts on *S. aureus* ATCC 43300 and *K. oxytoca*.** (A) AMtw3 on *K. oxytoca*. B) AMtf15 on *K. oxytoca*; (C) AMf6 on *K. oxytoca*; (D) AMf4 on *K. oxytoca*; (E) AMsb23 on *K. oxytoca*; (F) AMtw3 on *S. aureus*; (G) AMtf15 on *S.aureus* ATCC; (H) AMf6 on *S. aureus*; (I) AMf4 on *S. aureus*; (J) AMsb23 *on S. aureus*. AMf: *A. muricata* fruit, AMtf: *A. muricata* thorn of fruit, AMtw: *A. muricata* twigs, AMsb: *A. muricata* stem bark, MIC: Minimal Inhibitory Concentration; NC: Negative Control. Data are expressed as the mean±SD.

from 3.125 to 100 μg/mL depending on the extracts and microorganisms. The most active extracts against the three bacterial pathogens were consistently from the *Fusarium* and *Penicillium* genera, with *F. waltergamsii* AMtw3 being the most active, followed by *P. citrinum* AMf6. These findings agree with previous studies reporting the potential of endophytic fungi belonging to these genera as sources of active antimicrobial metabolites [15, 16]. Fortunately, these active endophyte extracts were noncytotoxic against Vero cells at concentrations as high as 100 μg/mL using the MTT colorimetric method. These results suggest that the antibacterial activity of extracts reported in the present study could not be related to the toxicity effect; therefore, antibacterial compounds produced by these endophytes could have good selectivity against mammalian cells. On the other hand, crude ethyl acetate extracts from endophytes belonging to these genera and inhabiting other Cameroonian medicinal plants were previously reported for their weak cytotoxicity against normal cell lines such as HEK239T mammalian

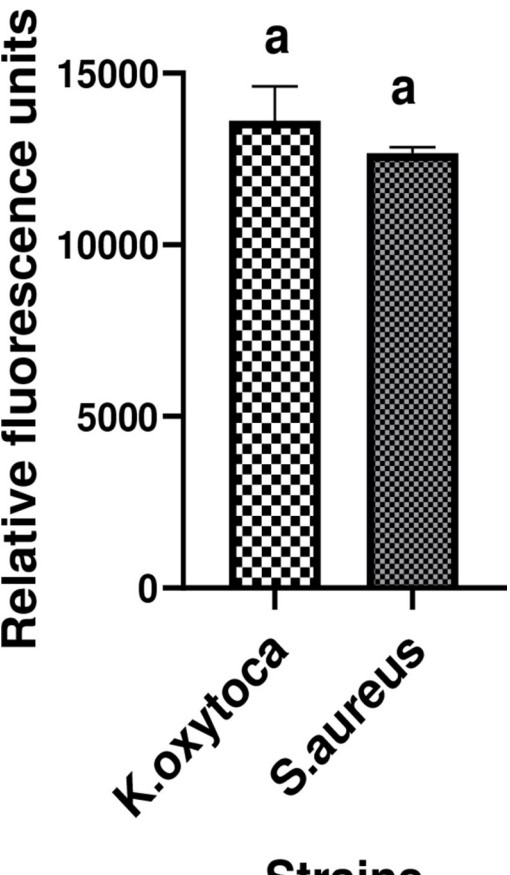

**Fig 5. Biomass of biofilms produced by K.oxytoca and S.aureus ATCC 43300 after 48hours in Nutrient broth (NB) medium supplemented with 2% glucose.** The experiment was performed twice in triplicate, and data are expressed as the mean±SD. Values with the same letter express no significant difference at p>0.05.

**Table 4. Inhibitory concentration 50 (IC$_{50}$) values of endophytic fungal extracts against *K. oxytoca* and *S. aureus* biofilm formation.**

| Fungal extracts | IC$_{50}$ (µg/mL± SD) | |
|---|---|---|
| | *K. oxytoca* | *S. aureus* ATCC 43300 |
| *F. waltergamsii* AMtw3 | 0.25±0.02[a] | 0.36±0.00[a] |
| *Aspergillus* sp. AMtf15 | 0.73±0.09[a] | 1.90±0.38[b] |
| *P. citrinum* AMf6 | 3.52±0.36[b] | 6.39±0.13[c] |
| *Curvularia* sp. AMf4 | 11.68±0.17[c] | 11.08±0.73[d] |
| *T. annesophieae* AMsb23 | 3.91±0.69[b] | 4.98±0.03[e] |
| Ciprofloxacin | 0.03±0.00[a] | 0.07±0.00[a] |

AMf: *A. muricata* fruit, AMtf: *A. muricata* thorn of fruit, AMtw: *A. muricata* twigs, AMsb: *A. muricata* stem bark, IC$_{50}$: Inhibitory Concentration 50; along the column, values carrying the same letter superscripts are not significantly different (p>0.05), and values carrying different letters are significantly different (P<0.05). The experiment was performed twice in duplicate.

cells [18, 36, 37]. However, this noncytotoxicity of crude metabolites produced by these endophytes could also be related to the culture conditions. We previously found that extracts from *Fusarium* sp. N240 and *Xylaria* sp. N120 cultured in potato dextrose broth were cytotoxic against HEK239T cells but were nontoxic when the fungi were grown in Czapek Dox medium [37].

Fungal extracts can exert their antimicrobial effect on bacteria via one or various mechanisms of action. Therefore, several model assays were used to investigate the mode of antibacterial action of the five most potent extracts (AMtw3, AMtf15, AMf6, AMf4, and AMsb23) against *S. aureus* ATCC43300 and *K. oxytoca*. The five extracts inhibited the production of catalase by both pathogens. The ability of bacteria to produce catalase contributes to their pathogenicity by detoxifying the oxygen-dependent microbicidal products of phagocytic cells. The large concentration of H$_2$O$_2$ may overwhelm an organism's defenses and may prove fatal to microorganisms [38]. Therefore, the loss of the ability to produce catalase by *S. aureus* and *K. oxytoca* indicates their inability to deactivate antimicrobials by catalase production [39]. These five extracts also demonstrated their ability to affect both the inner and outer membranes of bacteria. The results from the bacteriolysis assay indicate that extracts cause gross membrane

**Table 5. DPPH radical scavenging parameters of the seven active endophytic fungal extracts.**

| Extracts | RSA$_{50}$ (µg/mL± SD) | CE$_{50}$ | ARP |
|---|---|---|---|
| *Aspergillus* sp. AMtf15 | 663.35±4.73[a] | 3.31 x 10$^{4a}$ | 3.02 x 10$^{-5a}$ |
| *T. clemensii* AMtf5 | 146.05±4.31[b] | 0.73 x 10$^{4b}$ | 13.69 x 10$^{-5b}$ |
| *P. citrinum* AMf6 | 799.75±11.66[c] | 3.99 x 10$^{4c}$ | 2.50 x 10$^{-5c}$ |
| *F. waltergamsii* AMtw3 | 282.30±0.98[d] | 1.41 x 10$^{4d}$ | 7.09 x 10$^{-5d}$ |
| *Aspergillus* sp AMr9 | 176.90±0.84[e] | 0.88 x 10$^{4e}$ | 11.36 x 10$^{-5e}$ |
| *T. annesophieae* AMsb23 | 536.15±38.39[f] | 2.68 x 10$^{4f}$ | 3.73 x 10$^{-5e}$ |
| *P.citrinum* AMsb1 | 211.30±1.13[g] | 1.05 x 10$^{4g}$ | 9.52 x 10$^{-5f}$ |
| Vitamin C | 8.92±1.065[h] | 0.0446 x 10$^{4h}$ | 224.21 x 10$^{-5h}$ |

RSA$_{50}$: radical scavenging activity 50, EC$_{50}$: efficient concentration 50, ARP: antiradical power, NA: not applicable, AMr: *A. muricata* root, AMf: *A. muricata* fruit, AMtf: *A. muricata* thorn of fruit, AMtw: *A. muricata* twigs, AMsb: *A. muricata* stem bark. Along the column, values carrying the same letter superscripts are not significantly different (p>0.05), and values carrying different letters are significantly different (P<0.05).

**Table 6. Quantitative evaluation of $Fe^{3+}$ reducing power by endophytic extracts of *A. muricata*.**

| Code of extracts | μg equivalent $NH_2OH$/g of extract± SD | | | | | | |
|---|---|---|---|---|---|---|---|
| Concentrations (μg/mL) | 1000 | 500 | 250 | 125 | 62.5 | 31.25 | 15.625 |
| *Aspergillus* sp. AMtf15 | 12.03±0.59[a] | 11.46±0.16[ab] | 11.45±0.40[abc] | 11.36±0.34[bc] | 7.96±0.20[d] | 7.65±0.07[de] | 7.28±0.20[de] |
| *Penicillium* sp AMrb1 | 2.98±0.10[g] | 1.99±0.06[f] | 1.45±0.01[e] | 1.06±0.02[a] | 0.91±00[ab] | 0.79±0.2[bc] | 0.68±0.11[bc] |
| *P. citrinum* AMf6 | 12.28±0.36[a] | 11.88±0.12[ab] | 11.77±0.43[abc] | 11.72±0.38[abc] | 8.25±0.50[d] | 7.88±0.09[de] | 7.70±0.07[de] |
| *P.citrinum* AMs3 | 11.97±0.36[a] | 11.70±0.16[ab] | 11.57±0.16[abc] | 11.48±0.06[abc] | 7.84±0.09[e] | 4.12±0.35[d] | 3.76±0.43[d] |
| *Meyerozyma* sp.AMf3 | 1.06±0.00[a] | 0.95±0.08[ab] | 0.84±0.03[abc] | 0.83±0.06[abcd] | 0.82±0.06[abcde] | 0.79±0.10[abcdef] | 0.67±0.01[abcdef] |
| *Penicillium* sp. AMrb11 | 1.06±0.02[a] | 0.87±0.06[ab] | 0.82±0.19[abc] | 0.79±0.11[ab]c | 0.74±0.21[abcde] | 0.74±0.16[abcde] | 0.65±0.03[abcde] |
| *Curvularia* sp.AMf4 | 12.35±0.42[a] | 11.74±0.25[ab] | 11.53±1.06[abc] | 11.30±0.74[abcd] | 11.15±0.95[abcde] | 10.28±0.00[abcdef] | 8.47±0.43[f] |
| *Talaromyces* sp.AMtf5 | 3.92±0.18[e] | 2.67±0.08[d] | 1.76±0.08[c] | 1.26±0.03[a] | 0.95±0.07[ab] | 0.84±0.02[b] | 0.81±0.13[b] |
| *Aspergillus* sp.AMb7 | 1.10±0.00[a] | 1.08±0.01[ab] | 0.87±0.02[abc] | 0.80±0.02[abcd] | 0.76±0.00[bcde] | 0.73±0.06[cdef] | 0.69±0.20[cdef] |
| *Aspergillus* sp AMsb23 | 4.47±0.26[f] | 2.86±0.05[e] | 1.71±0.07[a] | 1.32±0.28[ab] | 1.01±0.15[abc] | 0.95±0.20[bcd] | 0.81±0.14[bcd] |
| *F. waltergamsii* AMtw3 | 2.26±0.00[e] | 1.61±0.01[d] | 1.41±0.02[a] | 1.29±0.06[a] | 1.08±0.06[b] | 1.01±0.01[b] | 0.69±0.02[c] |
| *P.citrinum*AMf1 | 12.37±0.02[a] | 11.83±0.16[ab] | 11.72±0.25[abc] | 11.70±0.22[abc] | 8.11±0.37[d] | 7.95±0.17[d] | 4.35±0.32[e] |
| *Aspergillus* sp AMr9 | 3.79±0.19[f] | 2.68±0.12[e] | 1.86±0.00[d] | 1.36±0.09[a] | 1.15±0.05[ab] | 0.89±0.03[abc] | 0.89±0.19[abc] |
| *Aspergillus* sp.AMrb9 | 1.09±0.01[e] | 0.94±0.00[d] | 0.84±0.03[a] | 0.81±0.00[a] | 0.74±0.01[b] | 0.68±0.00[bc] | 0.67±0.00[c] |
| *Aspergillus* sp AMsb1 | 4.90±0.26[e] | 3.27±0.13a | 2.20±0.09[a] | 1.57±0.03[ab] | 1.2±0.07[abc] | 1.06±0.22[abcd] | 0.79±0.00[acd] |
| *Aspergillus* sp.AMr10 | 1.43±0.03[f] | 1.20±0.00[a] | 1.16±0.02[ab] | 1.05±0.11[abc] | 0.95±0.02[bcd] | 0.91±0.05[cde] | 0.72±0.02[e] |

AMr: *A. muricata* root, AMrb: *A. muricata* root bark, AMf: *A. muricata* fruit, AMs: *A. muricata* seed, AMtf: *A. muricata* thorn of fruit, AMtw: *A. muricata*, AMb: *A. muricata* bark, AMsb: *A. muricata* stem bark; along the line, values carrying the same letter are not significantly different (P>0.05), and values carrying different letters are significantly different (P<0.05).

damage and provoke whole-cell lysis. This property of endophytic fungal extracts was previously reported by Mbekou *et al.* [23]. The outer membrane permeability assay results showed a gradual decrease in optical densities with increasing extract concentration, which indicates membrane destabilization. Lethal injury of microbial cell membranes may alter their permeability and affect the membrane's ability to osmoregulate the cell adequately or exclude toxic materials [40]. This antibacterial action can result in membrane expansion, increased membrane fluidity and permeability, disturbance of membrane-embedded proteins, inhibition of respiration, and alterations in the ionic homeostasis between intracellular and extracellular compartments of bacteria, eventually leading to cell death [41].

All five fungal extracts also exert a bactericidal effect on both pathogens as materialised by reducing the bacterial population, as shown by the graph of optical densities versus incubation time. Bactericidal agents are sought after to fight against resistant bacteria because they attack and kill bacteria outright, preventing these cells from causing further damage within the body [42]. Among bacterial therapeutic targets, biofilms remain the main virulence factor contributing to the pathogenesis and resistance of microbial pathogens [4]. Therefore, finding antibiotics with the ability to inhibit both planktonic cells and bacterial biofilm structure is of paramount importance. Our investigation revealed that extracts from AMtw3, AMtf15, AMf6, AMf4, and AMsb23 strongly inhibited *S. aureus* ATCC43300 and *K. oxytoca* biofilm formation ($IC_{50}$ 0.25 to 11.86 μg/mL). The inhibition of biofilm formation by extracts from endophytic fungi has also been reported by Dawande *et al.* [43] and Kaur *et al.* [44]. The antibiofilm-guided investigation of these extracts could identify very potent antibiofilm metabolites, a potential starting point for new drug discovery.

The potential of extracts to prevent oxidative stress resulting from excess nonneutralized reactive species in the body was also investigated through DPPH radical scavenging and FRAP reducing power assays. Antioxidants are tremendously important substances that can protect

the body from damage caused by free radicals. The scavenging activity of the fungal extracts was measured by discoloration to yellow following the formation of a nonradical (2,2-diphenyl-1-hydrazine) molecule [45]. Fungal extracts AMtf15, AMf6, AMtw3, AMr9, AMsb1, AMsb23 and AMf4 exhibited good DPPH radical scavenging activity (RSA$_{50}$ 146.05 to 799.75 µg/mL). The metal chelating capacity may also indicate its potential antioxidant activity [46]. The iron-chelating activity of all fungal extracts was determined by reaction with ortho-phenanthroline. The results showed that five extracts at the highest concentration tested, including AMtf15, AMf6, AMs3, AMf4, and AMf1, exhibited iron-chelating activity (12.03, 12.28, 11.97, 12.35 and 12.37 µg equivalent $NH_2OH$/g, respectively). Overall, two extracts from *Aspergillus* sp. AMtf15 and *P. citrinum* AMf6 exhibited DPPH radical scavenging and ferric ion reducing capacities. The antioxidant activities of these fungal extracts observed in the present study could be related to their phytochemical content. Our previous compositional analysis of ethyl acetate extracts from endophytic fungi from *Annona muricata*, *Aspergillus* sp. AMb7 and *P. citrinum* AMrb23 revealed the presence of 44 and 38 different secondary metabolites, with isolongifolene (30.18%) and octadecyl-3,5-di-tert-butyl-4-hydroxydrocinnamate (19.62%) being the most abundant in AMb7 and AMrb23, respectively [18]. Therefore, endophytic fungi from *Annona muricata* can produce a significant number of metabolites with the potential to act as both antimicrobials and antioxidants.

## Conclusion

The present study was designed to provide insight into the antibacterial, antioxidant and mode of action of crude metabolites from endophytic fungi inhabiting *Annona muricata* growing in Cameroon. Our investigation showed that a high percentage of isolates (41.46%) belonging to six different genera (*Aspergillus*, *Curvularia*, *Fusarium*, *Meyerozyma*, *Penicillium*, and *Talaromyces*) exhibited potent antibacterial activity against the causative agent of UTIs. The more potent displayed various modes of antibacterial action and antioxidant activity. These results suggest that each of the active endophytes identified from the present study can produce antibacterial molecules. Further antibacterial-guided fractionation is currently ongoing to purify and identify active compounds that may serve as a starting point for developing new pharmacological agents.

## Supporting information

**S1 File.**
(DOCX)

## Acknowledgments

The authors are highly grateful to Mr. Victor Nana for plant identification and specimen archiving. To Centre Pasteur du Cameroun (CPC) for providing us with the isolate of *Klebsiella oxytoca*. We also thank Manuel Sanchez Hernandez from the DNA Sequencing Service, University of Salamanca, Spain.

## Author Contributions

**Conceptualization:** Rufin Marie Kouipou Toghueo, Fabrice Fekam Boyom.

**Data curation:** Lorette Victorine Yimgang, Rufin Marie Kouipou Toghueo.

**Formal analysis:** Lorette Victorine Yimgang, Rufin Marie Kouipou Toghueo.

**Funding acquisition:** Fabrice Fekam Boyom.

**Investigation:** Lorette Victorine Yimgang, Ines Michele Kanko Mbekou, Darline Dize.

**Methodology:** Lorette Victorine Yimgang, Ines Michele Kanko Mbekou, Darline Dize.

**Project administration:** Rufin Marie Kouipou Toghueo, Fabrice Fekam Boyom.

**Resources:** Rufin Marie Kouipou Toghueo, Fabrice Fekam Boyom.

**Software:** Lorette Victorine Yimgang.

**Supervision:** Rufin Marie Kouipou Toghueo, Fabrice Fekam Boyom.

**Validation:** Lorette Victorine Yimgang, Rufin Marie Kouipou Toghueo.

**Visualization:** Rufin Marie Kouipou Toghueo, Fabrice Fekam Boyom.

**Writing – original draft:** Lorette Victorine Yimgang, Rufin Marie Kouipou Toghueo.

**Writing – review & editing:** Rufin Marie Kouipou Toghueo, Fabrice Fekam Boyom.

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
