## [Decision Letter · Decision Letter 0]

21 Mar 2022

PONE-D-22-00760Crude metabolites from endophytic fungi inhabiting Cameroonian Annona muricata exhibit potent inhibition against the causative agents of urinary tract infections (UTIs)PLOS ONE

Dear Dr. tOGHUEO,

Thank you for submitting your manuscript to PLOS ONE. After careful consideration, we feel that it has merit but does not fully meet PLOS ONE’s publication criteria as it currently stands. Therefore, we invite you to submit a revised version of the manuscript that addresses the points raised during the review process.

We look forward to receiving your revised manuscript.

Kind regards,

Guadalupe Virginia Nevárez-Moorillón, Ph.D.

Academic Editor

PLOS ONE

Journal Requirements:

Reviewers' comments:

Reviewer's Responses to Questions

**Comments to the Author**

1. Is the manuscript technically sound, and do the data support the conclusions?

Reviewer #1: Yes

Reviewer #2: Yes

2. Has the statistical analysis been performed appropriately and rigorously? 

Reviewer #1: Yes

Reviewer #2: Yes

3. Have the authors made all data underlying the findings in their manuscript fully available?

Reviewer #1: Yes

Reviewer #2: Yes

4. Is the manuscript presented in an intelligible fashion and written in standard English?

Reviewer #1: No

Reviewer #2: Yes

5. Review Comments to the Author

Reviewer #1: In this paper, the authors present a group of 41 endophytic fungi extracts from different parts of the Cameroonian plant Annona muricata and their effect on Staphylococcus aureus, Escherichia coli, and Klebsiella oxytoca. Seventeen extracts showed antibacterial effect and were selected for sequencing and identification by BLAST, six different fungi were identified. Finally, the five most active extracts were selected to evaluate their mode of action.

The study is well conceived and conducted, and the results appear to be solid; however, some considerations for improving the work are suggested.

Title. Staphylococcus aureus, Escherichia coli, and Klebsiella oxytoca are not specific bacteria of Urinary Tract Infections it would be convenient to change the more general title without specifying the disease, for example: "Crude metabolites from endophytic fungi inhabiting Cameroonian Annona muricata exhibit potent antibacterial effect".

Methodology. Homogenize brand and country of reagents and equipment.

Methodology: In the section "Antibacterial screening and minimum inhibitory concentrations of endophytes", the authors describe the evaluation process, but the active or non-active interpretation strategy is not clear, was it by turbidity?

Line 138. Correct parentheses

Line 184-185. Change "hydrogen peroxide" to "hydrogen peroxide (H2O2)".

Line 210: Is the concentration "5.10 7 CFU/mL" correct? Or does it refer to "1.5x10 7 CFU/mL"?

Line 382: Change the sentence "All extracts exhibited activity against all tested bacteria..." because the AMr10 extract only had activity against one bacterium.

Results. The results and figures of the effect of extracts on the permeability of outer cell membrane indicate "seven different concentrations", however, the methodology indicates three concentrations "MIC, 2MIC, and 4 MIC".

Growth curve. In Figure 4, the optical density is 600 nm and in the results and methodology, it is described at 620 nm, which is correct?

Line 490. (IC50 0.24-0.36 μg/mL) and (IC50, 0.73-490 1.903 μg/mL), in each case refers to two different concentrations and is confused with ranges; change to (IC50 0.24 and 0.36 μg/mL) and (IC50, 0.73 and 1.903 μg/mL),

The discussion does not mention the significance of the cytotoxicity result in Vero cells.

Correct the size of the "MIC" symbology in Fig. 2B.

The metabolites of F. waltergamsii and P. citrinum AMf6 were the most active, it is important to emphasize the discussion in this regard.

In figure legends change "S. Aureus" by "S. aureus"

Reviewer #2: The paper entitled " Crude metabolites from endophytic fungi inhabiting Cameroonian Annona muricata exhibit potent inhibition against the causative agents of urinary tract infections (UTIs): provide a comprehensive and significant view of the research area; I think the study is relevant and suitable for publication to the journal; however, there are some shortcoming and grammatical mistakes

> It is my hope that my comments will help the authors improve their manuscript and is the intention.

> One minor comment is that the title seems a little lengthy, please try a make it short and attractive

2nd minor comment is that there are some writing details in the manuscript.

>In title you haven’t mention “isolation” please revise it.

Page, line 22, put comma after “method”

Page, line 25, replace “Both the” to “The”

Line 27, MIC After first appearance it should be abbreviate, In abstract it’s already mentioned.

Page, line 40, “content” to “contain”

Page line 47, put comma after “infection”

Page, line 49, replace “spectre” to “specter”

Page, line 59, delete the word “being”

Page, line 60, replace the word “of”

Page, line 87, remove the comma after “antibacterial”

Line 91 and 318, Please revise it Use same pattern.

145, put comma after “mg/mL”

Line 153, Used same wording 24 h (Twenty-four) Please revise it whole manuscript.

Page, line 158-159 the endophytic fungi producing more potent secondary metabolites, revise this sentence.

Page, Line 161 Three bacterial species in every sentence you have used “THE” I think it should be “Extracts were screened against three bacterial species at single concentration” as well as

162 line should also revise.

Line 162 Mentioned CLSI after first appear, In Line 670 it should be abbreviate.

169, line 169, delete the word “that were”

Line 170, replace “in order to” to “to”

Line 171, replace “For the determination of” to “to determine”

Line 173, delete “as”

Line 174, replace “where there was” to “with”

Line 176, put comma after “1%”

Line 179, put comma after “AMf4”

Line 181, please revise it.

Line 192, put comma after “buffer”

Line 204, delete the word “respectively”

Line 214, delete “graph of” and rewrite “graph” after “OD/450 nm)”

Line 215, replace “was” to “were”

Line 227, Please revise it. As well as Revise line 231.

Line 233, put comma after “solution”

Line 270, put comma after “min”

Line 289, replace “was treated in” to “treated”

Line 326 revise it.

Line 399, put comma after “AMrb9”

Line 399, replace “rootbark” to “root bark”

Line 466, replace “that of” to “than of”

Page, line 508, write “was” before “obtained respectively”

Page, line 516, replace “have substantial influence on” to “substantially influence”

Page, line 523, delete the word “for the ability” as well as Revise it, above in manuscript it’s already mentioned 17, so used same pattern.

Page, line 581, write that before “over”

Page, line 605 replace “defences” to “defenses” Please revise it

Page, line 625, delete the word “with the ability”

Page, line 629, replace “antibiofilm guided” to “antibiofilm-guided”

Page, line 639, replace “serve as a significant indicator of” to “significantly indicate”

Page, line 642, put comma after “AMf1”

Page, line 659, replace “mode” to modes”

6. PLOS authors have the option to publish the peer review history of their article (what does this mean?). If published, this will include your full peer review and any attached files.

Reviewer #1: No

Reviewer #2: No

---

## [Author Response · Author response to Decision Letter 0]

3 Apr 2022

Response to Reviewers

Reviewer 1

Title. Staphylococcus aureus, Escherichia coli, and Klebsiella oxytoca are not specific bacteria of Urinary Tract Infections it would be convenient to change the more general title without specifying the disease, for example: "Crude metabolites from endophytic fungi inhabiting Cameroonian Annona muricata exhibit potent antibacterial effect". 

Response: Thank you very much for your suggestion. We chose the current title because this project aimed to target UTIs' causative agents specifically. Although Staphylococcus aureus, Escherichia coli, and Klebsiella oxytoca are not only involved in Urinary Tract Infections, they are among the most common agents of UTIs, accounting for about 75-80% of the case (Flores-Mireles et al., 2015). Besides, these species are among the most isolated from patients suffering from UTIs in Cameroon and other African countries (Akoachere et al., 2012; Bitew et al., 2017). 

Supporting literatures

Akoachere, JF.T.K., Yvonne, S., Akum, N.H. et al. Etiologic profile and antimicrobial susceptibility of community-acquired urinary tract infection in two Cameroonian towns. BMC Res Notes 5, 219 (2012). https://doi.org/10.1186/1756-0500-5-219

Bitew, A., Molalign, T. & Chanie, M. Species distribution and antibiotic susceptibility profile of bacterial uropathogens among patients complaining urinary tract infections. BMC Infect Dis 17, 654 (2017). https://doi.org/10.1186/s12879-017-2743-8

Flores-Mireles, A. L., Walker, J. N., Caparon, M., & Hultgren, S. J. (2015). Urinary tract infections: epidemiology, mechanisms of infection and treatment options. Nature reviews. Microbiology, 13(5), 269–284. https://doi.org/10.1038/nrmicro3432

Methodology. Homogenize brand and country of reagents and equipment. 

Response: We have now homogenized all brand and country names of reagents and equipment 

Methodology: In the section "Antibacterial screening and minimum inhibitory concentrations of endophytes", the authors describe the evaluation process, but the active or non-active interpretation strategy is not clear, was it by turbidity? 

Response: Yes, the turbidity was the strategy used to define the activity of potent extracts.

Line 138. Correct parentheses. 

Response: Thank you, parentheses are now corrected

Line 184-185. Change "hydrogen peroxide" to "hydrogen peroxide (H2O2)". 

Response: hydrogen peroxide is now changed to "hydrogen peroxide (H2O2)"

Line 210: Is the concentration "5.10 7 CFU/mL" correct? Or does it refer to "1.5x10 7 CFU/mL"? 

Response: Thank you for the question. The appropriate concentration was 5x107 CFU/mL. we have now edited the methodology accordingly

Line 382: Change the sentence "All extracts exhibited activity against all tested bacteria..." because the AMr10 extract only had activity against one bacterium. 

Response: The sentence "All extracts exhibited activity against all tested bacteria..." is now replaced by "All extracts exhibited activity against at least one of the tested bacteria with the MIC values ranging from 3.125 to 100 μg/mL depending on the extracts and microorganisms. "

Results. The results and figures of the effect of extracts on the permeability of outer cell membrane indicate "seven different concentrations", however, the methodology indicates three concentrations "MIC, 2MIC, and 4 MIC". 

Response: Thank you very much for this remark. Yes indeed, we tested 7 concentrations for the outer cell membrane permeability assays including 1/16 MIC, 1/8 MIC, 1/4 MIC, 1/2 MIC, MIC, 2MIC, and 4 MIC. The methodology is now revised. Thank you!

Growth curve. In Figure 4, the optical density is 600 nm and in the results and methodology, it is described at 620 nm, which is correct? 

Response: The right optical density is 620nm. We have now edited the figure accordingly.

Line 490. (IC50 0.24-0.36 μg/mL) and (IC50, 0.73-490 1.903 μg/mL), in each case refers to two different concentrations and is confused with ranges; change to (IC50 0.24 and 0.36 μg/mL) and (IC50, 0.73 and 1.903 μg/mL), 

Response: we have now changed (IC50 0.24-0.36 μg/mL) and (IC50, 0.73-490 1.903 μg/mL) to (IC50 0.24 and 0.36 μg/mL) and (IC50, 0.73 and 1.903 μg/mL).

The discussion does not mention the significance of the cytotoxicity result in Vero cells. Response: Thank you for this remark. We have now included the discussion on the significance of the cytotoxicity result in Vero cells.

Correct the size of the "MIC" symbology in Fig. 2B. 

Response: we have now harmonized the size of the "MIC" symbology in all figures 

The metabolites of F. waltergamsii and P. citrinum AMf6 were the most active, it is important to emphasize the discussion in this regard. 

Response: Thank you for your suggestion. We mentioned in the discussion that these findings agreed with previous reports. References 15 and 16 cited in the discussion are two comprehensive reviews articles detailing the biological activity of metabolites produced by endophytic fungi from the Fusarium and Penicillium genera. Therefore, we feel that these review articles are sufficient to support our findings allowing us to avoid unnecessary repetition of information.

Supporting information: 

15) Toghueo, 2020: Bioprospecting endophytic fungi from Fusarium genus as sources of bioactive metabolites. doi:10.1080/21501203.2019.1645053

16) Toghueo and Boyom, 2020: Endophytic Penicillium species and their agricultural, biotechnological, and pharmaceutical applications. doi:10.1007/s13205-020-2081-1

In figure legends change "S. Aureus" by "S. aureus".

Response: We have now replaced "S. Aureus" by "S. aureus"

2. Reviewer 2

One minor comment is that the title seems a little lengthy, please try a make it short and attractive

Response: Thank you for the suggestion. We have now revised the title.

2nd minor comment is that there are some writing details in the manuscript.

>In title you haven’t mention “isolation” please revise it.

Response: Thank you for the remark. The word isolation was replaced by “inhabiting”, which means live within. In this context, both words have identical meanings. We choose to use the less common.

Page, line 22, put comma after “method”. 

Response: we have now put comma after “method”.

Page, line 25, replace “Both the” to “The”. 

Response: we have now replace “Both the” to “The”.

Line 27, MIC After first appearance it should be abbreviate, In abstract it’s already mentioned. Response: we have now revised the entire manuscript accordingly

Page, line 40, “content” to “contain”. 

Response: we have now replaced “content” with “contain”.

Page line 47, put comma after “infection”. 

Response: we have now put comma after “infection”.

Page, line 49, replace “spectre” to “specter”. 

Response: we have now replaced “spectre” to “specter”.

Page, line 59, delete the word “being”. 

Response: we have now revised the sentence.

Page, line 60, replace the word “of” .

Response: we have now revised the sentence.

Page, line 87, remove the comma after “antibacterial”. 

Response: we have now revised the sentence.

Line 91 and 318, Please revise it Use same pattern. 

Response: we have now revised the sentence.

145, put comma after “mg/mL”. 

Response: we have now revised the sentence.

Line 153, Used same wording 24 h (Twenty-four) Please revise it whole manuscript. 

Response: we have now revised the sentence.

Page, line 158-159 the endophytic fungi producing more potent secondary metabolites, revise this sentence. 

Response: we have now revised the sentence.

Page, Line 161 Three bacterial species in every sentence you have used “THE” I think it should be “Extracts were screened against three bacterial species at single concentration” as well as. Response: we have now revised the sentence.

162 line should also revise. 

Response: we have now revised the sentence.

Line 162 Mentioned CLSI after first appear, In Line 670 it should be abbreviate. Response: we have now revised the sentence.

169, line 169, delete the word “that were”. 

Response: we have now revised the sentence.

Line 170, replace “in order to” to “to”

Response: we have now replaced “in order to” to “to”

Line 171, replace “For the determination of” to “to determine”. 

Response: we have now revised the sentence.Line 173, delete “as”

Line 174, replace “where there was” to “with”. 

Response: we have now revised the sentence.

Line 176, put comma after “1%”

Response: we have now revised the sentence.

Line 179, put comma after “AMf4”.

Response: we have now revised the sentence. 

Line 181, please revise it. 

Response: we have now revised the sentence.

Line 192, put comma after “buffer”. 

Response: we have now revised the sentence.

Line 204, delete the word “respectively”

Response: we have now revised the sentence.

Line 214, delete “graph of” and rewrite “graph” after “OD/450 nm)”. 

Response: we have now revised the sentence.

Line 215, replace “was” to “were”

Response: we have now replaced “was” by “were”.

Line 227, Please revise it. As well as Revise line 231.

Response: we have now revised the sentence

Line 233, put comma after “solution”

Line 270, put comma after “min”.

Response: we have now put comma after “min”. 

Line 289, replace “was treated in” to “treated”

Response: we have now replaced “was treated in” by “treated”

Line 326 revise it. 

Response: we have now revised the sentence

Line 399, put comma after “AMrb9”.

Response: we have now put comma after “AMrb9”. Line 399, replace “rootbark” to “root bark”

Response: we have now replaced “rootbark” by “root bark”

Line 466, replace “that of” to “than of”. 

Response: we have now replaced “that of” by “than of”. 

Page, line 508, write “was” before “obtained respectively”

Response: we have now revised the sentence

Page, line 516, replace “have substantial influence on” to “substantially influence” 

Response: we have now revised the sentence

Page, line 523, delete the word “for the ability” as well as Revise it, above in manuscript it’s already mentioned 17, so used same pattern. 

Response: we have now revised the sentence

Page, line 581, write that before “over”. 

Response: we have now revised the sentence

Page, line 605 replace “defences” to “defenses” Please revise it. 

Response: we have now replaced “defences” by “defenses” 

Page, line 625, delete the word “with the ability”

Response: we have now deleted the word “with the ability”

Page, line 629, replace “antibiofilm guided” to “antibiofilm-guided”. 

Response: we have now replaced “antibiofilm guided” by “antibiofilm-guided”.

Page, line 639, replace “serve as a significant indicator of” to “significantly indicate”. 

Response: we have now revised the sentence

Page, line 642, put comma after “AMf1”. 

Response: we have now revised the sentence 

Page, line 659, replace “mode” to modes”. 

Response: we have now revised the sentence

---

## [Editor Report · Decision Letter 1]

6 Apr 2022

Crude metabolites from endophytic fungi inhabiting Cameroonian Annona muricata exhibit potent inhibition against the causative agents of urinary tract infections (UTIs)

PONE-D-22-00760R1

Dear Dr. tOGHUEO,

We’re pleased to inform you that your manuscript has been judged scientifically suitable for publication and will be formally accepted for publication once it meets all outstanding technical requirements.

Kind regards,

Guadalupe Virginia Nevárez-Moorillón, Ph.D.

Academic Editor

PLOS ONE
---

## [Editor Report · Acceptance letter]

8 Apr 2022

PONE-D-22-00760R1 

Crude metabolites from endophytic fungi inhabiting Cameroonian *Annona muricata* inhibit the causative agents of urinary tract infections 

Dear Dr. Kouipou Toghueo:

I'm pleased to inform you that your manuscript has been deemed suitable for publication in PLOS ONE. Congratulations! Your manuscript is now with our production department. 

Kind regards, 

on behalf of

Dr. Guadalupe Virginia Nevárez-Moorillón 

Academic Editor

PLOS ONE